# Faster Local Solvers for Graph Diffusion Equations

**Jiahe Bai** [1]    **Baojian Zhou** [1,2*]    **Deqing Yang** [1,2]    **Yanghua Xiao** [2]

[1] the School of Data Science, Fudan University,

[2] Shanghai Key Laboratory of Data Science, School of Computer Science, Fudan University

jhbai20,bjzhou,yangdeqing,shawyh@fudan.edu.cn

## Abstract

Efficient computation of graph diffusion equations (GDEs), such as Personalized PageRank, Katz centrality, and the Heat kernel, is crucial for clustering, training neural networks, and many other graph-related problems. Standard iterative methods require accessing the whole graph per iteration, making them time-consuming for large-scale graphs. While existing *local solvers* approximate diffusion vectors through heuristic local updates, they often operate sequentially and are typically designed for specific diffusion types, limiting their applicability. Given that diffusion vectors are highly localizable, as measured by the *participation ratio*, this paper introduces a novel framework for approximately solving GDEs using a *local diffusion process*. This framework reveals the suboptimality of existing local solvers. Furthermore, our approach effectively localizes standard iterative solvers by designing simple and provably sublinear time algorithms. These new local solvers are highly parallelizable, making them well-suited for implementation on GPUs. We demonstrate the effectiveness of our framework in quickly obtaining approximate diffusion vectors, achieving up to a hundred-fold speed improvement, and its applicability to large-scale dynamic graphs. Our framework could also facilitate more efficient *local message-passing* mechanisms for GNNs.

## 1   Introduction

Graph diffusion equations (GDEs), such as Personalized PageRank (PPR) [41, 44, 62], Katz centrality (Katz) [46], Heat kernel (HK) [21], and Inverse PageRank (IPR) [51], are fundamental tools for modeling graph data. These score vectors for nodes in a graph capture various aspects of their importance or influence. They have been successfully applied to many graph learning tasks including local clustering [2, 81], detecting communities [49], semi-supervised learning [78, 80], node embeddings [63, 67], training graph neural networks (GNNs) [10, 16, 19, 31, 75], and many other applications [32]. Specifically, given a propagation matrix $M$ associated with an undirected graph $\mathcal{G}(\mathcal{V}, \mathcal{E})$, a general graph diffusion equation is defined as

$$\boldsymbol{f} \triangleq \sum_{k=0}^{\infty} c_k \boldsymbol{M}^k \boldsymbol{s}, \tag{1}$$

where $\boldsymbol{f}$ is the diffusion vector computed from a source vector $\boldsymbol{s}$, and the sequence of coefficients $c_k$ satisfies $c_k \geq 0$. Equation (1) represents a system of linear, constant-coefficient ordinary differential equation, $\dot{\boldsymbol{x}}(t) = \boldsymbol{M}\boldsymbol{x}(t)$, with an initial condition $\boldsymbol{x}(0)$ and $t \geq 0$. Standard solvers [34, 57] for computing $\boldsymbol{f}$ require access to matrix-vector product operations $\boldsymbol{M}\boldsymbol{x}$, typically involving $\mathcal{O}(m)$ operations, where $m$ is the total number of edges in $\mathcal{G}$. This could be time-consuming when dealing with large-scale graphs [42].

---

[*]Corresponding author

38th Conference on Neural Information Processing Systems (NeurIPS 2024).

A key property of $\boldsymbol{f}$ is the high localization of its entry magnitudes, which reside in a small portion of $\mathcal{G}$. Figure 1 demonstrates this localization property of $\boldsymbol{f}$ on PPR, Katz, and HK. Leveraging this locality property allows for more efficient approximation using *local iterative solvers*, which heuristically avoid $\mathcal{O}(m)$ operations. Local push-based methods [2, 7, 49] or their variants [4, 12] are known to be closely related to Gauss-Seidel [49, 18]. However, current local push-based methods are fundamentally sequential iterative solvers and focused on specific types such as PPR or HK. These disadvantages limit their applicability to modern GPU architectures and their generalizability.

By leveraging the locality of $\boldsymbol{f}$, we propose, for the first time, a general local iterative framework for solving GDEs using a *local diffusion process*. A novel component of our framework is to model *local diffusion* as a *locally evolving set process* inspired by its stochastic counterpart [58]. We use this framework to localize commonly used standard solvers, demonstrating faster local methods for approximating $\boldsymbol{f}$. For example, the local gradient descent is simple, provably sublinear, and highly parallelizable. It can be accelerated further by using local momentum. **Our contributions are**

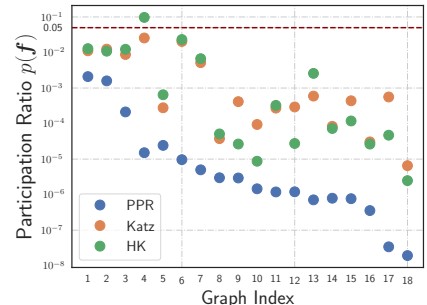

Figure 1: The maximal participation ratio $p(\boldsymbol{f}) = (\sum_{i=1}^{n} |f_i|^2)^2 / (n \sum_{i=1}^{n} |f_i|^4)$ of example diffusion vectors $\boldsymbol{f}$ over 18 graphs, ordered from small (*cora*) to large (*ogbn-papers100M*). The ratio $p(\boldsymbol{f})$ is normalized by the number of nodes $n$.

– By demonstrating that popular diffusion vectors, such as PPR, Katz, and HK, have strong localization properties using the participation ratio, we propose a novel graph diffusion framework via a *local diffusion process* for efficiently approximating GDEs. This framework effectively tracks most energy during diffusion while maintaining local computation.

– To demonstrate the power of our proposed framework, we prove that APPR [2], a widely used local push-based algorithm, can be treated as a special case. We provide better diffusion-based bounds $\widetilde{\Theta}(\overline{\text{vol}}(\mathcal{S}_t)/(\alpha \cdot \overline{\gamma}_t))$ where $\overline{\text{vol}}(\mathcal{S}_t)/\overline{\gamma}_t$ is a lower bound of $1/\epsilon$. This bound is effective for both PPR and Katz and is empirically smaller than $\Theta(1/(\alpha\epsilon))$, previously well-known for APPR.

– We design simple and fast local methods based on standard gradient descent for APPR and Katz, which admit runtime bounds of $\min(\overline{\text{vol}}(\mathcal{S}_t)/(\alpha \cdot \overline{\gamma}_t), 1/(\alpha\epsilon))$ for both cases. These methods are GPU-friendly, and we demonstrate that this iterative solution is significantly faster on GPU architecture compared to APPR. When the propagation matrix $\boldsymbol{M}$ is symmetric, we show that this local method can be accelerated further using the local Chebyshev method for PPR and Katz.

– Experimental results on GDE approximation of PPR, HK, and Katz demonstrate that these local solvers significantly accelerate their standard counterparts. We further show that they can be naturally adopted to approximate dynamic diffusion vectors. Our experiments indicate that these local methods for training are twice as fast as standard PPR-based GNNs. These results may suggest a novel *local message-passing* approach to train commonly used GNNs.

*All proofs, detailed experimental settings, and related works are postponed to the appendix. Our code is publicly available at https://github.com/JiaheBai/Faster-Local-Solver-for-GDEs.*

## 2 GDEs, Localization, and Existing Local Solvers

Table 1: Example GDEs with their corresponding propagation matrix $\boldsymbol{M}$, coefficients $c_k$, and source $\boldsymbol{s}$.

| Equ. | $\boldsymbol{M}$ | $c_k$ | $\boldsymbol{s}$ |
|------|------------------|-------|------------------|
| PPR [62] | $\boldsymbol{A}\boldsymbol{D}^{-1}$ | $\alpha(1-\alpha)^k$ | $\boldsymbol{e}_s$ |
| Katz [46] | $\boldsymbol{A}$ | $\alpha^k$ | $\boldsymbol{e}_s$ |
| HK [21] | $\boldsymbol{A}\boldsymbol{D}^{-1}$ | $e^{-\tau}\tau^k/k!$ | $\boldsymbol{e}_s$ |
| IPR [51] | $\boldsymbol{A}\boldsymbol{D}^{-1}$ | $\frac{\theta^k}{(\theta^k+\theta^{10})^2}$ | $\boldsymbol{e}_s$ |
| APPNP [30] | $\boldsymbol{D}^{-\frac{1}{2}}\boldsymbol{A}\boldsymbol{D}^{-\frac{1}{2}}$ | $\alpha(1-\alpha)^k$ | $\boldsymbol{x}$ |

**Notations.** We consider an undirected graph $\mathcal{G}(\mathcal{V}, \mathcal{E})$ where $\mathcal{V} = \{1, 2, \ldots, n\}$ is the set of nodes, and $\mathcal{E}$ is the edge set with $|\mathcal{E}| = m$. The degree matrix is $\boldsymbol{D} = \mathbf{diag}(d_1, d_2, \ldots, d_n)$, and $\boldsymbol{A}$ is the adjacency matrix of $\mathcal{G}$. The *volume* of subset nodes $\mathcal{S}$ is $\text{vol}(\mathcal{S}) = \sum_{v \in \mathcal{S}} d_v$. The set of *neighbors* of node $v$ is denoted by $\mathcal{N}(v)$. The standard basis for node $s$ is $\boldsymbol{e}_s$. The support of $\boldsymbol{x}$ is defined as $\text{supp}(\boldsymbol{x}) = \{u : x_u \neq 0\}$.

Many graph learning tools can be represented as diffusion vectors. Table 1 presents examples widely used as graph learning tools. The coefficient $c_k$ usually exhibits exponential decay, so most of the energy of $\boldsymbol{f}$ is related to only the first few $c_k$. We revisit these GDEs and existing local solvers.

**Personalized PageRank.** Given the weight decaying strategy $c_k = \alpha(1 - \alpha)^k$ and a predefined damping factor $\alpha \in (0, 1)$ with $\boldsymbol{M} = \boldsymbol{A}\boldsymbol{D}^{-1}$, the analytic solution for PPR is

$$\boldsymbol{f}_{\text{PPR}} = \alpha(\boldsymbol{I} - (1 - \alpha)\boldsymbol{A}\boldsymbol{D}^{-1})^{-1}\boldsymbol{e}_s. \tag{2}$$

These vectors can be used to find a local graph cut or train GNNs [8, 10, 25, 79]. The well-known approximate PPR (APPR) method [2] locally updates the estimate-residual pair $(\boldsymbol{x}, \boldsymbol{r})$ as follows

APPR : $\qquad \boldsymbol{x} \leftarrow \boldsymbol{x} + \alpha r_u \cdot \boldsymbol{e}_u, \qquad \boldsymbol{r} \leftarrow \boldsymbol{r} - r_u \cdot \boldsymbol{e}_u + (1 - \alpha)r_u \cdot \boldsymbol{A}\boldsymbol{D}^{-1} \cdot \boldsymbol{e}_u.$

Here, each *active* node $u$ has a large residual $r_u \geq \epsilon\alpha d_u$ with initial values $\boldsymbol{x} \leftarrow 0$ and $\boldsymbol{r} \leftarrow \alpha\boldsymbol{e}_s$. The runtime bound for reaching $r_u < \epsilon\alpha d_u$ for all nodes is graph-independent and is $\Theta(1/(\alpha\epsilon))$, leveraging the monotonicity property. Previous analyses [49, 18] suggest that APPR is a localized version of the Gauss-Seidel (GS) iteration and thus has fundamental sequential limitations.

**Katz centrality.** Given the weight $c_k = \alpha^{k+1}$ with $\alpha \in (0, 1/\|\boldsymbol{A}\|_2)$ and a different propagation matrix $\boldsymbol{M} = \boldsymbol{A}$. The solution for Katz centrality can then be rewritten as:

$$\boldsymbol{f}_{\text{Katz}} = (\boldsymbol{I} - \alpha\boldsymbol{A})^{-1}\boldsymbol{e}_s. \tag{3}$$

The goal is to approximate the Katz centrality $((\boldsymbol{I} - \alpha\boldsymbol{A})^{-1} - \boldsymbol{I})\boldsymbol{e}_s$. Similar to APPR, a local solver [11] for Katz centrality, denoted as AKatz, can be designed as follows

AKatz : $\qquad \boldsymbol{x} \leftarrow \boldsymbol{x} + r_u\boldsymbol{e}_u, \qquad \boldsymbol{r} \leftarrow \boldsymbol{r} - r_u\boldsymbol{e}_u + \alpha r_u\boldsymbol{A}\boldsymbol{e}_u.$

AKatz is a coordinate descent method [11]. Under the nonnegativity assumption of $\boldsymbol{r}$ (i.e., $\alpha \leq 1/d_{\max}$), a convergence bound for the residual $\|\boldsymbol{r}^{t+1}\|_1$ is provided.

**Heat Kernel.** The heat kernel (HK) [21] is useful for finding smaller but more precise local graph cuts. It uses a different weight decaying $c_k = \tau^k e^{-\tau}/k!$ and the vector is then defined as

$$\boldsymbol{f}_{\text{HK}} = \exp\left\{-\tau\left(\boldsymbol{I} - \boldsymbol{M}\right)\right\}\boldsymbol{e}_s,$$

where $\tau$ is the temperature parameter of the HK equation and $\boldsymbol{M} = \boldsymbol{A}\boldsymbol{D}^{-1}$. Unlike the previous two, this equation does not admit an analytic solution. Following the technique developed in [48, 49], given an initial vector $\boldsymbol{s}$ and temperature $\tau$, the HK vector can be approximated using the first $N$ terms of the Taylor polynomial approximation: define $\boldsymbol{x}_N = \sum_{k=0}^{N} \boldsymbol{v}_k$ with $\boldsymbol{v}_0 = \boldsymbol{e}_s$ and $\boldsymbol{v}_{k+1} = \boldsymbol{M}\boldsymbol{v}_k/k$ for $k = 0, \ldots, N$. It is known that $\|\boldsymbol{f}_{\text{HK}} - \boldsymbol{x}_N\|_1 \leq 1/(N!N)$. This is equivalent to solving the linear system $(\boldsymbol{I}_{N+1} \otimes \boldsymbol{I}_n - \boldsymbol{S}_{N+1} \otimes \boldsymbol{M})\boldsymbol{v} = \boldsymbol{e}_1 \otimes \boldsymbol{e}_s$, where $\otimes$ denotes the Kronecker product. Then, the push-based method, namely AHK, has the following updates

AHK : $\quad \boldsymbol{v} \leftarrow \boldsymbol{v} + r_u\left(\boldsymbol{e}_k \otimes \boldsymbol{e}_u\right), \qquad \boldsymbol{r} \leftarrow \boldsymbol{r} - r_u\left(\boldsymbol{I}_{N+1} \otimes \boldsymbol{I}_n - \boldsymbol{S}_{N+1} \otimes \boldsymbol{M}\right)\left(\boldsymbol{e}_k \otimes \boldsymbol{e}_u\right).$

There are many other types of GDEs, such as Inverse PageRank [51], which generalize PPR. Many GNN propagation layers, such as SGC [69] and APPNP [30], can be formulated as GDEs. For all these $\boldsymbol{f}$, we use the *participation ratio* [54] to measure its localization ability, defined as

$$p(\boldsymbol{f}) = \left(\sum_{i=1}^{n} |f_i|^2\right)^2 \Big/ \left(n \cdot \sum_{i=1}^{n} |f_i|^4\right).$$

When the entries in $\boldsymbol{f}$ are uniformly distributed, such that $f_i \sim \mathcal{O}(1/n)$, then $p(\boldsymbol{f}) = \mathcal{O}(1)$. However, in the extremely sparse case where $\boldsymbol{f}$ is a standard basis vector, $p(\boldsymbol{f}) = 1/n$ indicates the sparsity effect. Figure 1 illustrates the participation ratios, showing that almost all vectors have ratios below 0.01. Additionally, larger graphs tend to have smaller participation ratios.

APPR, AKatz, and AHK all have fundamental limitations due to their reliance on sequential Gauss-Seidel-style updates, which limit their parallelization ability. In the following sections, we will develop faster local methods using the local diffusion framework for solving PPR and Katz.

# 3 Faster Solvers via Local Diffusion Process

We introduce a local diffusion process framework, which allows standard iterative solvers to be effectively localized. By incorporating this framework, we show that the computation of PPR and Katz defined in (2) and (3) can be locally approximated sequentially or in parallel on GPUs.

## 3.1 Local diffusion process

To compute $\boldsymbol{f}_{\text{PPR}}$ and $\boldsymbol{f}_{\text{KATZ}}$, it is equivalent to solving the linear systems, which can be written as

$$\boldsymbol{Q}\boldsymbol{x} = \boldsymbol{s}, \tag{4}$$

where $\boldsymbol{Q}$ is a (symmetric) positive definite matrix with eigenvalues bounded by $\mu$ and $L$, i.e., $\mu \leq \lambda(\boldsymbol{Q}) \leq L$. For PPR, we solve $(\boldsymbol{I} - (1-\alpha)\boldsymbol{A}\boldsymbol{D}^{-1})\boldsymbol{x} = \alpha\boldsymbol{e}_s$, which is equivalent to solving $(\boldsymbol{I} - (1-\alpha)\boldsymbol{D}^{-1/2}\boldsymbol{A}\boldsymbol{D}^{-1/2})\boldsymbol{D}^{-1/2}\boldsymbol{x} = \alpha\boldsymbol{D}^{-1/2}\boldsymbol{e}_s$. For Katz, we compute $(\boldsymbol{I} - \alpha\boldsymbol{A})\boldsymbol{x} = \boldsymbol{e}_s$. The vector $\boldsymbol{s}$ is sparse, with $\text{supp}(\boldsymbol{s}) \ll n$. We define *local diffusion process*, a locally evolving set procedure inspired by its stochastic counterpart [58] as follows

**Definition 3.1** (Local diffusion process). Given an input graph $\mathcal{G}$, a source distribution $\boldsymbol{s}$, precision tolerance $\epsilon$, and a local iterative method $\mathcal{A}$ with parameter $\theta$ for solving (4), the local diffusion process is defined as a process of updates $\{(\boldsymbol{x}^{(t)}, \boldsymbol{r}^{(t)}, \mathcal{S}_t)\}_{0 \leq t \leq T}$. Specifically, it follows the dynamic system

$$\left(\boldsymbol{x}^{(t+1)}, \boldsymbol{r}^{(t+1)}, \mathcal{S}_{t+1}\right) = \phi\left(\boldsymbol{x}^{(t)}, \boldsymbol{r}^{(t)}, \mathcal{S}_t; \boldsymbol{s}, \epsilon, \mathcal{G}, \mathcal{A}_\theta\right), \quad 0 \leq t \leq T. \tag{5}$$

where $\phi$ is a mapping via some iterative solver $\mathcal{A}_\theta$. For all three diffusion processes (PPR, Katz, and HK), we set $\mathcal{S}_0 = \{s\}$. We say this process *converges* when $\mathcal{S}_T = \emptyset$ if there exists such $T$; the generated sequence of active nodes are $\mathcal{S}_t$. The total number of operations of the local solver $\mathcal{A}_\theta$ is

$$\mathcal{T}_{\mathcal{A}_\theta} = \sum_{t=0}^{T-1} \text{vol}(\mathcal{S}_t) = T \cdot \overline{\text{vol}}(\mathcal{S}_T),$$

where we denote the average of active volume as $\overline{\text{vol}}(\mathcal{S}_T) = T^{-1}\sum_{t=0}^{T-1}\text{vol}(\mathcal{S}_t)$.

Intuitively, we can treat this local diffusion process as a random walk on a Markov Chain defined on the space of all subsets of $\mathcal{V}$, where the transition probability represents the operation cost. More efficient local solvers try to find a *shorter* random walk from $\mathcal{S}_0$ to $\mathcal{S}_T$. The following two subsections demonstrate the power of this process in designing local methods for PPR and Katz.

## 3.2 Sequential local updates via Successive Overrelaxation (SOR)

Given the estimate $\boldsymbol{x}^{(t)}$ at $t$-th iteration, we define residual $\boldsymbol{r}^{(t)} = \boldsymbol{s} - \boldsymbol{Q}\boldsymbol{x}^{(t)}$ and $\boldsymbol{Q}$ be the matrix induced by $\boldsymbol{A}$ and $\boldsymbol{D}$. The typical local Gauss-Seidel with Successive Overrelaxation has the following online estimate-residual updates (See Section 11.2 of [34]): at each time $t = 0, 1, \ldots, T-1$, for each active node $u_i \in \mathcal{S}_t = \{u_1, u_2, \ldots, u_{|\mathcal{S}_t|}\}$ with $i = 1, 2, \ldots, |\mathcal{S}_t|$, we update each $u_i$-th entry of $\boldsymbol{x}$ and corresponding residual $\boldsymbol{r}$ as the following

$$\text{LocalSOR}: \quad \boldsymbol{x}^{(t+t_{i+1})} = \boldsymbol{x}^{(t+t_i)} + \omega \cdot \tilde{\boldsymbol{e}}_{u_i}^{(t+t_i)}, \quad \boldsymbol{r}^{(t+t_{i+1})} = \boldsymbol{r}^{(t+t_i)} - \omega \cdot \boldsymbol{Q} \cdot \tilde{\boldsymbol{e}}_{u_i}^{(t+t_i)}, \tag{6}$$

where $t_i \triangleq (i-1)/|\mathcal{S}_t|$ is the current time, and update unit vector is $\tilde{\boldsymbol{e}}_{u_i}^{(t+t_i)} \triangleq r_{u_i}^{(t+t_i)}\boldsymbol{e}_{u_i}/q_{u_iu_i}$. Note that each update of (6) is $\Theta(d_{u_i})$. If $\mathcal{S}_t = \mathcal{V}$, it reduces to the standard GS-SOR [34]. Therefore, APPR is a special case of (6), a local variant of GS-SOR with $\omega = 1$. Figure 2 illustrates this procedure. APPR updates $\boldsymbol{x}$ at some entries and keeps track of large magnitudes of $\boldsymbol{r}$ per iteration, while LocalSOR allows for a better choice of $\omega$; hence, the total number of operations is reduced. We establish the following fundamental property of LocalSOR.

**Theorem 3.2** (Properties of local diffusion process via LocalSOR). *Let $\boldsymbol{Q} \triangleq \boldsymbol{I} - \beta\boldsymbol{P}$ where $\boldsymbol{P} \geq \boldsymbol{0}_{n \times n}$ and $P_{uv} \neq 0$ if $(u,v) \in \mathcal{E}$; 0 otherwise. Define maximal value $P_{\max} = \max_{u \in \mathcal{V}}\|\boldsymbol{P}\boldsymbol{e}_u\|_1$. Assume that $\boldsymbol{r}^{(0)} \geq \boldsymbol{0}$ is nonnegative and $P_{\max}, \beta$ are such that $\beta P_{\max} < 1$, given the updates of (6), then the local diffusion process of $\phi\left(\boldsymbol{x}^{(t)}, \boldsymbol{r}^{(t)}, \mathcal{S}_t; \boldsymbol{s}, \epsilon, \mathcal{G}, \mathcal{A}_\theta = (\text{LocalSOR}, \omega)\right)$ with $\omega \in (0,1)$ has the following properties*

1. ***Nonnegativity.*** $\boldsymbol{r}^{(t+t_i)} \geq \boldsymbol{0}$ *for all $t \geq 0$ and $t_i = (i-1)/|\mathcal{S}_t|$ with $i = 1, 2, \ldots, |\mathcal{S}_t|$.*

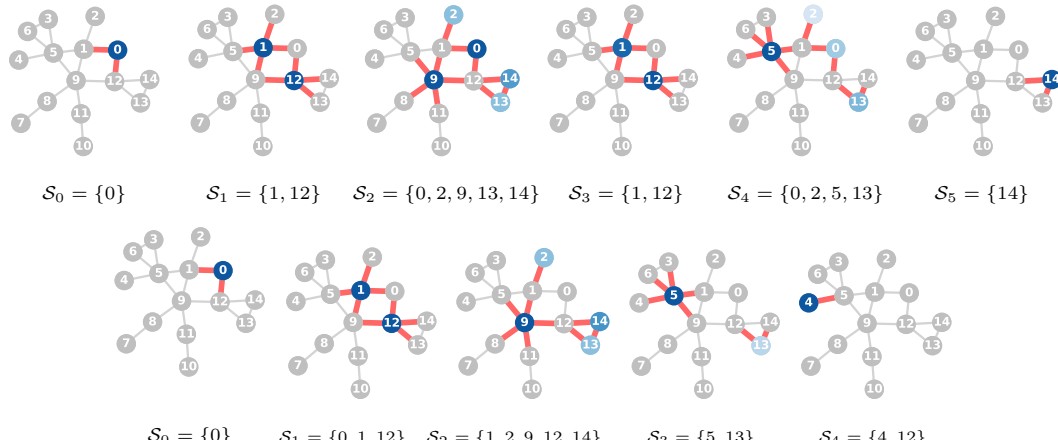

$$\mathcal{S}_0 = \{0\} \qquad \mathcal{S}_1 = \{1, 12\} \qquad \mathcal{S}_2 = \{0, 2, 9, 13, 14\} \qquad \mathcal{S}_3 = \{1, 12\} \qquad \mathcal{S}_4 = \{0, 2, 5, 13\} \qquad \mathcal{S}_5 = \{14\}$$

$$\mathcal{S}_0 = \{0\} \qquad \mathcal{S}_1 = \{0, 1, 12\} \qquad \mathcal{S}_2 = \{1, 2, 9, 12, 14\} \qquad \mathcal{S}_3 = \{5, 13\} \qquad \mathcal{S}_4 = \{4, 12\}$$

Figure 2: The first row illustrates the local diffusion process of APPR [2] over a toy network topology adopted from [40]. It uses $T_{\text{APPR}} = 6$ local iterations with $\mathcal{T}_{\text{APPR}} = 42$ operations and additive error $\approx 0.29241$. The second row shows the process of LocalSOR ($\omega = 1.19 \approx \omega^*$). It uses $T_{\text{LocalSOR}} = 5$ local iterations with $\mathcal{T}_{\text{LocalSOR}} = 28$ operations and additive error $\approx 0.21479$. LocalSOR uses fewer local iterations, costs less total active volume, and obtains better approximate solutions. We choose the source node $s = 0$ with $\epsilon = 0.02$ and $\alpha = 0.25$.

2. **Monotonicity property.** $\|\boldsymbol{r}^{(0)}\|_1 \geq \cdots \|\boldsymbol{r}^{(t+t_i)}\|_1 \geq \|\boldsymbol{r}^{(t+t_{i+1})}\|_1 \cdots$.

*If the local diffusion process converges (i.e., $\mathcal{S}_T = \emptyset$), then $T$ is bounded by*

$$T \leq \frac{1}{\omega \overline{\gamma}_T (1 - \beta P_{\max})} \ln \frac{\|\boldsymbol{r}^{(0)}\|_1}{\|\boldsymbol{r}^{(T)}\|_1}, \text{ where } \overline{\gamma}_T \triangleq \frac{1}{T} \sum_{t=0}^{T-1} \left\{ \gamma_t \triangleq \frac{\sum_{i=1}^{|\mathcal{S}_t|} r_{u_i}^{(t+t_i)}}{\|\boldsymbol{r}^{(t)}\|_1} \right\}.$$

Based on Theorem 3.2, we establish the following sublinear time bounds of LocalSOR for PPR.

**Theorem 3.3** (Sublinear runtime bound of LocalSOR for PPR). *Let $\mathcal{I}_T = \text{supp}(\boldsymbol{r}^{(T)})$. Given an undirected graph $\mathcal{G}$ and a target source node $s$ with $\alpha \in (0, 1), \omega = 1$, and provided $0 < \epsilon \leq 1/d_s$, the run time of LocalSOR in Equ. (6) for solving $(\boldsymbol{I} - (1 - \alpha)\boldsymbol{A}\boldsymbol{D}^{-1})\boldsymbol{f}_{PPR} = \alpha \boldsymbol{e}_s$ with the stop condition $\|\boldsymbol{D}^{-1}\boldsymbol{r}^{(T)}\|_\infty \leq \alpha \epsilon$ and initials $\boldsymbol{x}^{(0)} = \boldsymbol{0}$ and $\boldsymbol{r}^{(0)} = \alpha \boldsymbol{e}_s$ is bounded as the following*

$$\mathcal{T}_{LocalSOR} \leq \min \left\{ \frac{1}{\epsilon \alpha}, \frac{\overline{\text{vol}}(\mathcal{S}_T)}{\alpha \overline{\gamma}_T} \ln \frac{C_{PPR}}{\epsilon} \right\}, \qquad \text{where } \frac{\overline{\text{vol}}(\mathcal{S}_T)}{\overline{\gamma}_T} \leq \frac{1}{\epsilon}. \tag{7}$$

*where $C_{PPR} = 1/((1 - \alpha)|\mathcal{I}_T|)$. The estimate $\boldsymbol{x}^{(T)}$ satisfies $\|\boldsymbol{D}^{-1}(\boldsymbol{x}^{(T)} - \boldsymbol{f}_{PPR})\|_\infty \leq \epsilon$.*

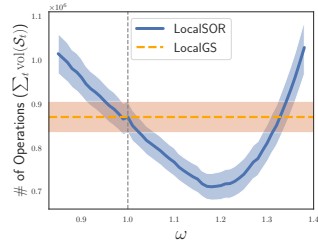

Figure 3: Parameter tuning of $\omega$.

The above theorem demonstrates the usefulness of our framework. It shows a new evolving bound where $\overline{\text{vol}}(\mathcal{S}_T)/\overline{\alpha}_T \leq 1/\epsilon$ as long as $\boldsymbol{Q}$ satisfying certain assumption (It is true for PPR and Katz). Similarly, applying Theorem 3.2, we have the following result for approximating $\boldsymbol{f}_{\text{Katz}}$.

**Corollary 3.4** (Runtime bound of LocalSOR for Katz). *Let $\mathcal{I}_T = \text{supp}(\boldsymbol{r}^{(T)})$ and $C_{Katz} = 1/((1 - \alpha)|\mathcal{I}_T|)$. Given an undirected graph $\mathcal{G}$ and a target source node $s$ with $\alpha \in (0, 1/d_{\max}), \omega = 1$, and provided $0 < \epsilon \leq 1/d_s$, the run time of LocalSOR in Equ. (6) for solving $(\boldsymbol{I} - \alpha\boldsymbol{A})\boldsymbol{x} = \boldsymbol{e}_s$ with the*

stop condition $\|\boldsymbol{D}^{-1}\boldsymbol{r}^{(T)}\|_\infty \leq \epsilon$ and initials $\boldsymbol{x}^{(0)} = \boldsymbol{0}$ and $\boldsymbol{r}^{(0)} = \boldsymbol{e}_s$ is bounded as the following

$$\mathcal{T}_{LocalSOR} \leq \min \left\{ \frac{1}{\epsilon (1 - \alpha d_{\max})}, \frac{\overline{\text{vol}}(\mathcal{S}_T)}{(1 - \alpha d_{\max})\overline{\gamma}_T} \ln \frac{C_{Katz}}{\epsilon} \right\}, \text{ where } \frac{\overline{\text{vol}}(\mathcal{S}_T)}{\overline{\gamma}_T} \leq \frac{1}{\epsilon}. \tag{8}$$

*The estimate $\hat{\boldsymbol{f}}_{Katz} = \boldsymbol{x}^{(T)} - \boldsymbol{e}_s$ satisfies $\|\hat{\boldsymbol{f}}_{Katz} - \boldsymbol{f}_{Katz}\|_2 \leq \|(\boldsymbol{I} - \alpha\boldsymbol{A})^{-1}\boldsymbol{D}\|_1 \cdot \epsilon$.*

**Accerlation when $\boldsymbol{Q}$ is Stieltjes matrix** ($\omega^* = 2/(1 + \sqrt{1 - (\alpha - 1)^2})$). It is well-known that when $\omega \in (1, 2]$, GS-SOR has acceleration ability when $\boldsymbol{Q}$ is Stieltjes. However, it is fundamentally

difficult to prove the runtime bound as the monotonicity property no longer holds. It has been conjectured that a runtime of $\tilde{\mathcal{O}}(1/(\sqrt{\alpha}\epsilon))$ can be achieved [27]. Incorporating our bound, a new conjecture could be $\tilde{\mathcal{O}}(\overline{\mathrm{vol}}(\mathcal{S}_T)/(\sqrt{\alpha}\overline{\gamma}_T))$: If $\boldsymbol{Q}$ is Stieltjes, then with the proper choice of $\omega$, one can achieve speedup convergence to $\tilde{\mathcal{O}}(\overline{\mathrm{vol}}(\mathcal{S}_T)/(\sqrt{\alpha}\overline{\alpha}_T))$? We leave it as an open problem.

## 3.3 Parallelizable local updates via GD and Chebyshev

The fundamental limitation of LocalSOR is its reliance on essentially sequential online updates, which may be challenging to utilize with GPU-type computing resources. *Interestingly, we developed a local iterative method that is embarrassingly simpler, highly parallelizable, and provably sublinear for approximating PPR and Katz.* First, one can reformulate the equation of PPR and Katz as [2]

$$\boldsymbol{x}_t^* = \arg\min_{\boldsymbol{x}\in\mathbb{R}^n} f(\boldsymbol{x}) \triangleq \frac{1}{2}\boldsymbol{x}^\top\boldsymbol{Q}\boldsymbol{x} - \boldsymbol{s}^\top\boldsymbol{x}, \tag{9}$$

A natural idea for solving the above is to use standard GD. Hence, following a similar idea of local updates, we simultaneously update solutions for $\mathcal{S}_t$. We propose the following local GD.

$$\text{LocalGD}: \qquad \boldsymbol{x}^{(t+1)} = \boldsymbol{x}^{(t)} + \boldsymbol{r}_{\mathcal{S}_t}^{(t)}, \qquad \boldsymbol{r}^{(t+1)} = \boldsymbol{r}^{(t)} - \boldsymbol{Q}\boldsymbol{r}_{\mathcal{S}_t}^{(t)} \tag{10}$$

**Theorem 3.5** (Properties of local diffusion process via LocalGD)**.** *Let* $\boldsymbol{Q} \triangleq \boldsymbol{I} - \beta\boldsymbol{P}$ *where* $\boldsymbol{P} \geq \boldsymbol{0}_{n\times n}$ *and* $P_{uv} \neq 0$ *if* $(u,v) \in \mathcal{E}$; *0 otherwise. Define maximal value* $P_{\max} = \max_{\mathcal{S}_t \subseteq \mathcal{V}} \|\boldsymbol{P}\boldsymbol{r}_{\mathcal{S}_t}\|_1 / \|\boldsymbol{r}_{\mathcal{S}_t}\|_1$. *Assume that* $\boldsymbol{r}^{(0)} = \boldsymbol{s} \geq \boldsymbol{0}$ *and* $P_{\max}, \beta$ *are such that* $\beta P_{\max} < 1$, *given the updates of* (10), *then the local diffusion process of* $\phi\left(\mathcal{S}_t, \boldsymbol{x}^{(t)}, \boldsymbol{r}^{(t)}; \mathcal{G}, \mathcal{A}_\theta = (\text{LocalGD}, \mu, L)\right)$ *has the following properties*

1. ***Nonnegativity.*** $\boldsymbol{r}^{(t)} \geq \boldsymbol{0}$ *for all* $t \geq 0$.

2. ***Monotonicity property.*** $\|\boldsymbol{r}^{(0)}\|_1 \geq \cdots \|\boldsymbol{r}^{(t)}\|_1 \geq \|\boldsymbol{r}^{(t+1)}\|_1 \cdots$.

*If the local diffusion process converges (i.e.,* $\mathcal{S}_T = \emptyset$*), then* $T$ *is bounded by*

$$T \leq \frac{1}{\overline{\gamma}_T(1-\beta P_{\max})} \ln\frac{\|\boldsymbol{r}^{(0)}\|_1}{\|\boldsymbol{r}^{(T)}\|_1}, \text{ where } \overline{\gamma}_T \triangleq \frac{1}{T}\sum_{t=0}^{T-1}\left\{\gamma_t \triangleq \frac{\|\boldsymbol{r}_{\mathcal{S}_t}^{(t)}\|_1}{\|\boldsymbol{r}^{(t)}\|_1}\right\}.$$

Indeed, the above local updates have properties that are quite similar to those of SOR. Based on Theorem 3.5, we establish the sublinear runtime bounds of LocalGD for solving PPR and Katz.

**Corollary 3.6** (Convergence of LocalGD for PPR and Katz)**.** *Let* $\mathcal{I}_T = \mathrm{supp}(\boldsymbol{r}^{(T)})$ *and* $C = \frac{1}{(1-\alpha)|\mathcal{I}_T|}$. *Use* LocalGD *to approximate PPR or Katz by using iterative procedure* (10). *Denote* $\mathcal{T}_{PPR}$ *and* $\mathcal{T}_{Katz}$ *as the total number of operations needed by using* LocalGD, *they can then be bounded by*

$$\mathcal{T}_{PPR} \leq \min\left\{\frac{1}{\alpha_{PPR}\cdot\epsilon}, \frac{\overline{\mathrm{vol}}(\mathcal{S}_T)}{\alpha_{PPR}\cdot\overline{\gamma}_T}\ln\frac{C}{\epsilon}\right\}, \quad \frac{\overline{\mathrm{vol}}(\mathcal{S}_T)}{\overline{\gamma}_T} \leq \frac{1}{\epsilon} \tag{11}$$

*for a stop condition* $\|\boldsymbol{D}^{-1}\boldsymbol{r}^{(t)}\|_\infty \leq \alpha_{PPR}\cdot\epsilon$. *For solving* KATZ*, then the toal runtime is bounded by*

$$\mathcal{T}_{Katz} \leq \min\left\{\frac{1}{(1-\alpha_{Katz}\cdot d_{\max})\epsilon}, \frac{\overline{\mathrm{vol}}(\mathcal{S}_T)}{(1-\alpha_{Katz}\cdot d_{\max})\overline{\gamma}_T}\ln\frac{C}{\epsilon}\right\}, \quad \frac{\overline{\mathrm{vol}}(\mathcal{S}_T)}{\overline{\gamma}_T} \leq \frac{1}{\epsilon} \tag{12}$$

*for a stop condition* $\|\boldsymbol{D}^{-1}\boldsymbol{r}^{(t)}\|_\infty \leq \epsilon d_u$. *The estimate of equality is the same as that of LocalSOR.*

*Remark* 3.7. Note that LocalGD is quite different from iterative hard-thresholding methods [43] where the time complexity of the thresholding operator is $\mathcal{O}(n)$ at best, hence not a sublinear algorithm.

**Accerlerated local Chevbyshev.** One can extend the Chebyshev method [39], an optimal iterative solver for (9). Following [24], Chebyshev polynomials $T_n$ are defined via following recursions

$$\mathcal{T}_t(x) = 2x\mathcal{T}_{t-1}(x) - \mathcal{T}_{t-2}(x), \quad \text{for } k \geq 2, \quad \text{with } \mathcal{T}_0(t) = 1, \mathcal{T}_1(t) = x. \tag{13}$$

---

[2]For the PPR equation, $(\boldsymbol{I} - (1-\alpha)\boldsymbol{A}\boldsymbol{D}^{-1})\boldsymbol{x} = \alpha\boldsymbol{e}_s$, we can symmetrize this linear system by rewriting it as $(\boldsymbol{I} - (1-\alpha)\boldsymbol{D}^{-1/2}\boldsymbol{A}\boldsymbol{D}^{-1/2})\boldsymbol{D}^{-1/2}\boldsymbol{x} = \alpha\boldsymbol{D}^{-1/2}\boldsymbol{e}_s$. The solution is then recovered by $\boldsymbol{D}^{1/2}\boldsymbol{x}^{(t)}$.

Given $\delta_1 = \frac{L-\mu}{L+\mu}, \boldsymbol{x}_1 = \boldsymbol{x}_0 - \frac{2}{L+\mu}\boldsymbol{\nabla}f(\boldsymbol{x}_0)$, the standard Chevyshev method is defined as

$$\boldsymbol{x}_k = \boldsymbol{x}_{k-1} - \frac{4\delta_k}{L-\mu}\nabla f(\boldsymbol{x}_{k-1}) + \left(1 - 2\delta_k\frac{L+\mu}{L-\mu}\right)(\boldsymbol{x}_{k-2} - \boldsymbol{x}_{k-1}), \delta_k = \frac{1}{2\frac{L+\mu}{L-\mu} - \delta_{k-1}},$$

where $\mu \le \lambda(\boldsymbol{Q}) \le L$. For example, for approximating PPR, we propose the following local updates

$$\text{LocalCH}: \boldsymbol{\pi}^{(t+1)} = \boldsymbol{\pi}^{(t)} + \frac{2\delta_{t+1}}{1-\alpha}\boldsymbol{D}^{1/2}\boldsymbol{r}_{\mathcal{S}_t}^{(t)} + \delta_{t:t+1}(\boldsymbol{\pi}^{(t)} - \boldsymbol{\pi}^{(t-1)})_{\mathcal{S}_t}, \delta_{t+1} = \left(\frac{2}{1-\alpha} - \delta_t\right)^{-1}$$
$$\boldsymbol{D}^{1/2}\boldsymbol{r}^{(t+1)} = \boldsymbol{D}^{1/2}\boldsymbol{r}^{(t)} - (\boldsymbol{\pi}^{(t+1)} - \boldsymbol{\pi}^{(t)}) + (1-\alpha)\boldsymbol{A}\boldsymbol{D}^{-1}(\boldsymbol{\pi}^{(t+1)} - \boldsymbol{\pi}^{(t)}).$$

The sublinear runtime analysis for LocalCH is complicated since it does not follow the monotonicity property during the updates. Whether it admits, an accelerated rate remains an open problem.

## 4 Applications to Dynamic GDEs and GNN Propagation

Our accelerated local solvers are ready for many applications. We demonstrate here that they can be incorporated to approximate dynamic GDEs and GNN propagation. We consider the *discrete-time dynamic graph* [47] which contains a sequence of snapshots of the underlying graphs, i.e., $\mathcal{G}_0, \mathcal{G}_1, \ldots, \mathcal{G}_T$. The transition from $\mathcal{G}_{t-1}$ to $\mathcal{G}_t$ consists of a list of events $\mathcal{O}_t$ involving edge deletions or insertions. The graph $\mathcal{G}_t$ is then represented as: $\mathcal{G}_0 \xrightarrow{\mathcal{O}_1} \mathcal{G}_1 \xrightarrow{\mathcal{O}_2} \mathcal{G}_2 \xrightarrow{\mathcal{O}_3} \cdots \mathcal{G}_{T-1} \xrightarrow{\mathcal{O}_T} \mathcal{G}_T$. The goal is to calculate an approximate $\boldsymbol{f}_t$ for the PPR linear system $\boldsymbol{Q}_t\boldsymbol{f}_t = \alpha\boldsymbol{e}_s$ at time $t$. That is,

$$\left(\boldsymbol{I} - (1-\alpha)\boldsymbol{A}_t\boldsymbol{D}_t^{-1}\right)\boldsymbol{f}_t = \alpha\boldsymbol{e}_s. \tag{14}$$

The key advantage of updating the above equation (presented in Algorithm 3) is that the APPR algorithm is used as a primary component for updating, making it much cheaper than computing from scratch. Consequently, we can apply our local solver LocalSOR to the above equation, and we found that it significantly sped up dynamic GDE calculations and dynamic PPR-based GNN training.

## 5 Experiments

**Datasets.** We conduct experiments on 18 graphs ranging from small-scale (*cora*) to large-scale (*papers100M*), mainly collected from Stanford SNAP [45] and OGB [42] (see details in Table 3). We focus on the following tasks: 1) approximating diffusion vectors $\boldsymbol{f}$ with fixed stop conditions using both CPU and GPU implementations; 2) approximating dynamic PPR using local methods and training dynamic GNN models based on InstantGNN models [75]. [3]

**Baselines and Experimental Setups.** We consider four methods with their localized counterparts: Gauss-Seidel (GS)/LocalGS, Successive Overrelaxation (SOR)/LocalSOR, Gradient Descent (GD)/LocalGD, and Chebyshev (CH)/LocalCH.[4] Specifically, we use the stop condition $\|\boldsymbol{D}^{-1}\boldsymbol{r}^{(t)}\|_\infty \le \epsilon\alpha$ for PPR and $\|\boldsymbol{D}^{-1}\boldsymbol{r}^{(t)}\|_\infty \le \epsilon$ for Katz, while we follow the parameter settings used in [49] for HK. For LocalSOR, we use the optimal parameter $\omega^*$ suggested in Section 3.2. We randomly sample 50 nodes uniformly from lower to higher-degree nodes for all experiments. All results are averaged over these 50 nodes. We conduct experiments using Python 3.10 with CuPy and Numba on a server with 80 cores, 256GB of memory, and two 28GB NVIDIA-4090 GPUs.

### 5.1 Results on efficiency of local GDE solvers

**Local solvers are faster and use fewer operations.** We first investigate the efficiency of the proposed local solvers for computing PPR and Katz centrality. We set $(\alpha_{\text{PPR}} = 0.1, \epsilon = 1/n)$ for PPR and $(\alpha_{\text{Katz}} = 1/(\|\boldsymbol{A}\|_2 + 1), \epsilon = 1/m)$ for Katz. We use a temperature of $\tau = 10$ and $\epsilon = 1/\sqrt{n}$ for HK. All algorithms for HK estimate the number of iterations by considering the truncated error of Taylor approximation. Table 2 presents the *speedup ratio* of four local methods over their standard counterparts. For five graph datasets, the speedup is more than 100 times in terms of the number of operations in most cases. This strongly demonstrates the efficiency of these local solvers. Figure

---

[3]More detailed experimental setups and additional results are in the appendix.

[4]Note that the approximate local solvers of APPR, AHK, and AKatz will be referred to as GS.

4 presents all such results. Key observations are: 1) *All local methods significantly speed up their global counterparts on all datasets.* This strongly indicates that when $\epsilon$ is within a certain range, local solvers for GDEs are much cheaper than their global counterparts. 2) *Among all local methods, LocalGS and LocalGD have the best overall performance.* This may seem counterintuitive since LocalSOR and LocalCH are more efficient in convergence rate. However, the number of iterations needed for $\epsilon = 1/n$ or $\epsilon = 1/m$ is much smaller. Hence, their improvements are less significant.

Table 2: Speedup ratio of computing PPR vectors. Let $\mathcal{T}_{\mathcal{A}}$ and $\mathcal{T}_{\text{Local}\mathcal{A}}$ be the number of operations of the standard algorithm $\mathcal{A}$ and the local solver Local$\mathcal{A}$, respectively. The speedup ratio $= \mathcal{T}_{\mathcal{A}}/\mathcal{T}_{\text{Local}\mathcal{A}}$.

| Graph | SOR/LocalSOR | GS/LocalGS | GD/LocalGD | CH/LocalCH |
|---|---|---|---|---|
| Citeseer | 86.21 | 114.89 | 157.41 | 5.86 |
| ogbn-arxiv | 183.43 | 392.26 | 528.03 | 101.88 |
| ogbn-products | 667.39 | 765.97 | 904.21 | 417.41 |
| wiki-talk | 169.67 | 172.95 | 336.53 | 30.45 |
| ogbn-papers100M | 243.68 | 809.47 | 1137.41 | 131.30 |

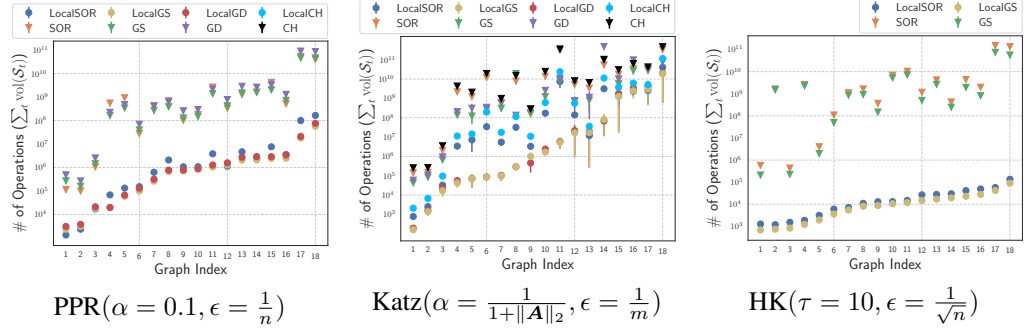

Figure 4: Number of operations required for four representative methods and their localized counterparts over 18 graphs. The graph index is sorted according to the performance of LocalGS.

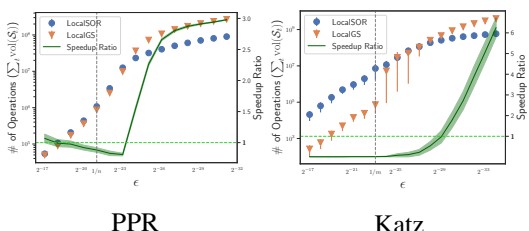

PPR      Katz

Figure 5: The number of operations as a function of $\epsilon$ for comparing LocalSOR and LocalGS.

**Which local GDE solvers are the best under what settings?** In the lower precision setting, previous results suggest that LocalSOR and LocalGD perform best overall. To test the proposed LocalSOR efficiency, we use the *ogbn-arxiv* dataset to evaluate the local algorithm efficiency under a high precision setting. Figure 5 illustrates the performance of LocalSOR and LocalGS for PPR and Katz. As $\epsilon$ becomes smaller, the speedup of LocalSOR over LocalGS becomes more significant.

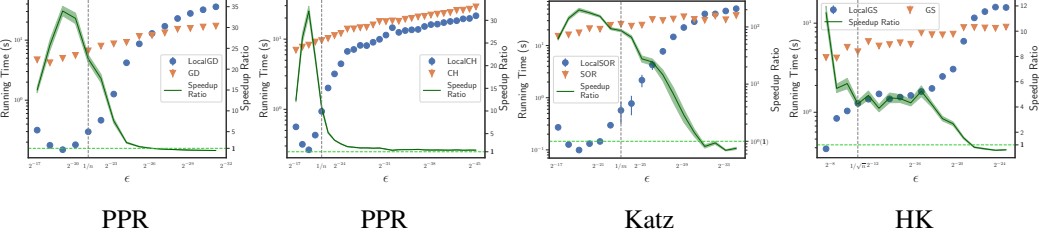

PPR      PPR      Katz      HK

Figure 6: Running time (seconds) as a function of $\epsilon$ for HK, Katz, and PPR GDEs on the *wiki-talk* dataset. We use $\alpha_{\text{PPR}} = 0.1$, $\alpha_{\text{Katz}} = 1/(\|\boldsymbol{A}\|_2 + 1)$, and $\tau_{\text{HK}} = 10$ with 50 sampled source nodes.

**When do local GDE solvers (not) work?** The next critical question is when standard solvers can be effectively localized, meaning under which $\epsilon$ conditions we see speedup over standard solvers. We conduct experiments to determine when local GDE solvers show speedup over their global counterparts for approximating Katz, HK, and PPR on the *wiki-talk* graph dataset for different $\epsilon$ values, ranging from lower precision ($2^{-17}$) to high precision ($2^{-32}$). As illustrated in Figure 6,

when $\epsilon$ is in the lower range, local methods are much more efficient than the standard counterparts. As expected, the speedup decreases and becomes worse than the standard counterparts when high precision is needed. This indicates the broad applicability of our framework; the required precisions in many real-world graph applications are within a range where our framework is effective.

## 5.2 Local GDE solvers on GPU-architecture

We conducted experiments on the efficiency of the GPU implementation of LocalGD and GD, specifically using LocalGD's GPU implementation to compare with other methods. Figure 7 presents the running time of global and local solvers as a function of $\epsilon$ on the *wiki-talk* dataset. LocalGD is the fastest among a wide range of $\epsilon$. This indicates that, when a GPU is available and $\epsilon$ is within the effective range, LocalGD (GPU) can be much faster than standard GD (GPU) and other methods based on CPUs. We observed similar patterns for computing Katz in Figure 11.

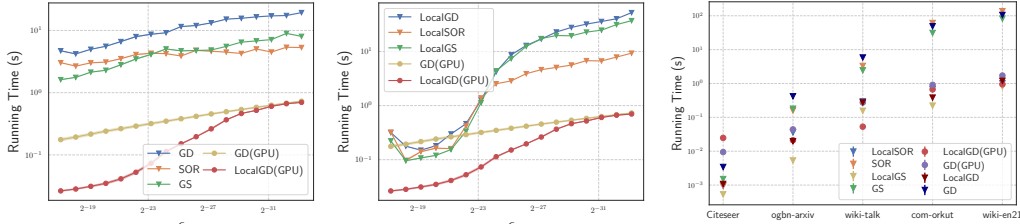

Figure 7: Comparison of running time (seconds) for CPU and GPU implementations.

## 5.3 Dynamic PPR approximating and training GNN models

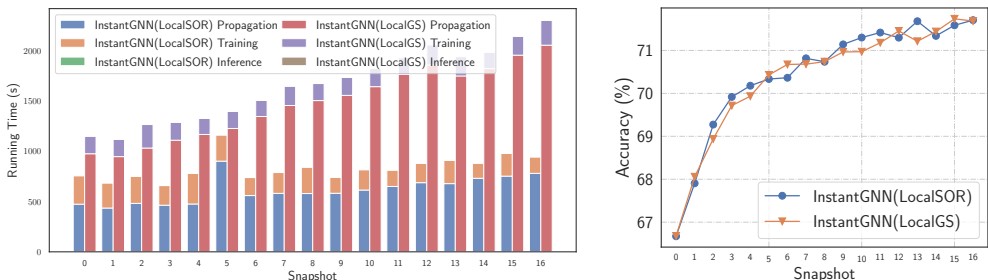

Figure 8: InstantGNN model using local iterative solver (LocalSOR) to do propagation.

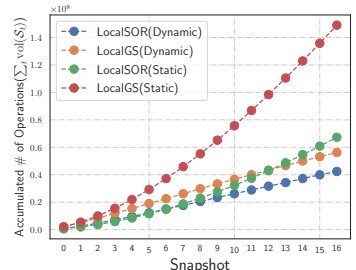

Figure 9: Acculmulated total number of operations of local solvers on the *ogbn-arxiv* dataset.

To test the applicability of the proposed local solvers, we apply LocalSOR to GNN propagation and test its efficiency on *ogbn-arxiv* and *ogbn-products*. We use the InstantGNN [75] model (essentially APPNP [30]), following the experimental settings in [75]. Specifically, we set $\alpha = 0.1$, $\epsilon = 10^{-2}/n$, and $\omega = \omega^*$ to the optimal value for our cases. For *ogbn-arxiv*, we randomly partition the graph into 16 snapshots (each snapshot with 59,375 edges), where the initial graph $\mathcal{G}_0$ contains 17.9% of the edges. With a similar performance on testing accuracy, as illustrated in Figure 8, the Instant-GNN model with LocalSOR (Algo. 7) has a significantly shorter training time than its local propagation counterpart (Algo. 6). Figure 9 presents the accumulated operations over these 16 snapshots. The faster local solver is LocalSOR (Dynamic), which dynamically updates $(\boldsymbol{x}, \boldsymbol{r})$ for (14) according to Algo. 5 and Algo. 3, whereas LocalSOR (Static) updates all approximate PPR vectors from scratch at the start of each snapshot for (14). We observed a similar pattern for LocalCH, where the number of operations is significantly reduced compared to static ones.

# 6 Limitations and Conclusion

When $\epsilon$ is sufficiently small, the speedup is insignificant, and it is unknown whether more efficient local solvers can be designed under this setting. Although we observed acceleration in practice, the accelerated bounds for LocalSOR and LocalCH have not been proven. Another limitation of local solvers is that they inherit the limitations of their standard counterparts. This paper mainly develops local solvers for PPR and Katz, and it remains interesting to consider other types of GDEs.

We propose the local diffusion process, a local iterative algorithm framework based on the locally evolving set process. Our framework is powerful in capturing existing local iterative solvers such as APPR for GDEs. We then demonstrate that standard iterative solvers can be effectively localized, achieving sublinear runtime complexity when monotonicity properties hold in these local solvers. Extensive experiments show that local solvers consistently speed up standard solvers by a hundredfold. We also show that these local solvers could help build faster GNN models [31, 13, 14, 20]. We expect many GNN models to benefit from these local solvers using a *local message passing* strategy, which we are actively investigating. Several open problems are worth exploring, such as whether these empirically accelerated local solvers admit accelerated sublinear runtime bounds without the monotonicity assumption.

## Acknowledgments and Disclosure of Funding

The authors would like to thank the anonymous reviewers for their helpful comments. The work of Baojian Zhou is sponsored by Shanghai Pujiang Program (No. 22PJ1401300) and the National Natural Science Foundation of China (No. KRH2305047). The work of Deqing Yang is supported by Chinese NSF Major Research Plan No.92270121. The computations in this research were performed using the CFFF platform of Fudan University.

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

# A Related Work

**Disclaimer**: The local diffusion process introduced in this paper shares a similar conceptual foundation with our concurrent work [77], where we propose the locally evolving set process. However, these two approaches address distinct problems. In this work, our focus is on investigating a different class of equations and exploring whether the local diffusion process can accelerate the training of Graph Neural Networks (GNNs).

**Graph diffusion equations (GDEs) and localization.** Graph diffusion vectors can either be exactly represented as a linear system or be approximations of an underlying linear system. The general form defined in Equation (1) has appeared in various literature [49, 51] (see more examples in [68]) or can be found in the survey works of [32]. The locality and localization properties of the diffusion vector $f$, including the PPR vector, have been explored in [64, 2, 59, 33, 60]. Inspired by methods for measuring the dynamic localization of quantum chaos [35, 22], which are also used for analyzing graph eigenfunctions [11, 26, 54], we use the *inverse participation ratio* to measure the localization ability of $f$. However, there is a lack of a local framework for solving GDEs efficiently.

**Standard solvers for GDEs.** The computation of a graph diffusion equation approximates the exponential of a propagation matrix, either through an inherently Neumann series or as an analytic system of linear equations. Well-established iterative methods, such as those approximating via the Neumann series or Padé approximations, exist (see more details in the surveys [57, 6, 61] and [34]). Each iteration requires computing the matrix-vector dot product, requiring $\mathcal{O}(m)$ operations for all the above methods. This is the computation we aim to avoid.

**Local methods for GDEs.** The most well-known local solver for solving PPR was initially proposed in [2] for local clustering. PPR, as a fundamental tool for graph learning, has been applied to many core problems, including graph neural networks [8, 17, 69, 15], graph diffusion [31, 74], and its generalizations [51, 19] for GNNs or clustering [50, 71]. Many variants have been proposed for solving such equations in different forms using local methods that couple local methods with Monte Carlo sampling strategies, achieving sublinear time [5, 53]. The work of [55] attempts to accelerate PPR computation via the Gauss-Southwell method, but it does not work well in practice [48]. A notable finding from [70] is that the local computation of PPR is equivalent to power iteration when the precision is high. There are also works on temporal PPR vector computation [66, 36, 56, 52], and our work can be immediately applied to these. Some local variants of APPR have been proposed for Katz and HK [11, 49]. However, these local solvers are sequential and lack the ability to be easily implemented on GPUs. Our local framework addresses this challenge.

**PPR on dynamic graphs.** Ranking on dynamic graphs has many applications [65], including motif clustering [71, 29], embedding on large-scale graphs [63, 23, 37, 38, 76, 18], and designing graph neural networks [75, 72]. The local computation via forward push (a.k.a. APPR algorithm [2]) or reverse push (a.k.a. reverse APPR [1]) is widely used in GNN propagation and bidirectional propagation algorithms [16, 28]. Our framework helps to design more efficient dynamic GNNs.

# B Missing Proofs

## B.1 Notations

We list all notations used in our proofs here.

- $e_u$: The indicator vector where the $u$-th entry is 1, and 0 otherwise.
- $\mathcal{N}(u)$: Set of neighbors of node $u$.
- $d_u$: The degree of node $u$.
- $\mathcal{G}(\mathcal{V}, \mathcal{E})$: The underlying graph with the set of nodes $\mathcal{V}$ and set of edges $\mathcal{E}$.
- $A$: Adjacency matrix of $\mathcal{G}$.
- $D$: Degree matrix of $\mathcal{G}$ when $\mathcal{G}$ is undirected.
- $\Pi$: Personalized PPR matrix, defined as $\Pi = \alpha \left( I - (1 - \alpha) A D^{-1} \right)^{-1}$.
- $\alpha_{\text{PPR}}$: Damping factor $\alpha_{\text{PPR}} \in (0, 1)$ for the PPR equation.
- $\alpha_{\text{Katz}}$: Parameter $\alpha_{\text{Katz}} \in (0, 1)$ for the Katz equation.

- $\epsilon$: The error tolerance.
- $\|\boldsymbol{D}^{-1}\boldsymbol{r}\|_\infty \leq \alpha_{\text{PPR}}\epsilon$: The stop condition for local solvers of PPR.
- $\|\boldsymbol{D}^{-1}\boldsymbol{r}\|_\infty \leq \epsilon$: The stop condition for local solvers of Katz.

## B.2 Formulation of PPR and Katz

In this section, we restate all theorems and present all missing proofs. We first restate the two target linear systems we aim to solve, as introduced in Section 3. Recall PPR is defined as $\left(\boldsymbol{I} - (1 - \alpha_{\text{PPR}})\boldsymbol{A}\boldsymbol{D}^{-1}\right)\boldsymbol{f}_{\text{PPR}} = \alpha_{\text{PPR}}\boldsymbol{e}_s$. It can be equivalently written as

$$\text{PPR:} \quad \boldsymbol{Q}\boldsymbol{x} = \boldsymbol{s}, \quad \boldsymbol{Q} = \left(\boldsymbol{I} - (1 - \alpha_{\text{PPR}})\boldsymbol{D}^{-1/2}\boldsymbol{A}\boldsymbol{D}^{-1/2}\right)\boldsymbol{x} = \alpha_{\text{PPR}}\boldsymbol{D}^{-1/2}\boldsymbol{e}_s, \quad (15)$$

where $\alpha_{\text{PPR}} \in (0, 1)$ and $\boldsymbol{x}^* = \alpha\boldsymbol{Q}^{-1}\boldsymbol{D}^{-1/2}\boldsymbol{e}_s$. Hence, $\boldsymbol{f}_{\text{PPR}} = \boldsymbol{D}^{1/2}\boldsymbol{x}^*$. The Katz centrality can be written as

$$\text{Katz:} \quad \boldsymbol{Q}\boldsymbol{x} = \boldsymbol{s}, \quad (\boldsymbol{I} - \alpha_{\text{Katz}}\boldsymbol{A})\boldsymbol{x} = \boldsymbol{e}_s, \quad (16)$$

where $\alpha_{\text{Katz}} \in (0, 1/\|\boldsymbol{A}\|_2)$. Then $\boldsymbol{f}_{\text{Katz}} = \boldsymbol{x}^* - \boldsymbol{e}_s$. [5]

**Proposition B.1** ([3, 53]). *Let $\mathcal{G}(\mathcal{V}, \mathcal{E})$ be an undirected graph and $u$ and $v$ be two vertices in $\mathcal{V}$. Denote $\boldsymbol{f}_s := \alpha(\boldsymbol{I} - (1 - \alpha)\boldsymbol{A}\boldsymbol{D}^{-1})\boldsymbol{e}_s$ as the PPR vector of a source node $s$. Then*

$$d_u \cdot f_u[v] = d_v \cdot f_v[u].$$

*Proof.* We directly follow the proof strategy in [53] for completeness. For path $P = \{s, v_1, v_2, \ldots, v_k, t\}$ in $\mathcal{G}$, we denote its length as $\ell(P)$ (here $\ell(P) = k + 1$), and define its reverse path to be $\bar{P} = \{t, v_k, \ldots, v_2, v_1, s\}$ — note that $\ell(P) = \ell(\bar{P})$. Moreover, we know that a random walk starting from $s$ traverses path $P$ with probability $\mathbb{P}[P] = \frac{1}{d_s} \cdot \frac{1}{d_{v_1}} \cdot \cdots \cdot \frac{1}{d_{v_k}}$, and thus, it is easy to see that we have

$$\mathbb{P}[P] \cdot d_s = \mathbb{P}[\bar{P}] \cdot d_t.$$

Now let $\mathcal{P}_{st}$ denote the paths in $G$ starting at $s$ and terminating at $t$. Then, we can re-write $f_s[t]$ as:

$$f_s[t] = \sum_{P \in \mathcal{P}_{st}} \alpha(1 - \alpha)^{\ell(P)}\mathbb{P}[P]$$

$$= \sum_{P \in \mathcal{P}_{st}} \alpha(1 - \alpha)^{\ell(P)}\frac{d_t}{d_s}\mathbb{P}[\bar{P}]$$

$$= \frac{d_t}{d_s} \sum_{\bar{P} \in \mathcal{P}_{ts}} \alpha(1 - \alpha)^{\ell(\bar{P})}\mathbb{P}[\bar{P}]$$

$$= \frac{d_t}{d_s} f_t[s],$$

$\square$

---

**Algo 1** APPR$(\mathcal{G}, \epsilon, \alpha, s)$ [2] adopted from [8]

---

1: $\boldsymbol{x} \leftarrow \boldsymbol{0}, \quad \boldsymbol{r} \leftarrow \alpha\boldsymbol{e}_s$
2: **while** $\exists u, r_u \geq \epsilon\alpha d_u$ **do**
3: $\quad \boldsymbol{r}, \boldsymbol{x} \leftarrow \text{PUSH}(u, \boldsymbol{r}, \boldsymbol{x})$
4: **Return** $\boldsymbol{x}$
1: $\text{PUSH}(u, \boldsymbol{r}, \boldsymbol{x})$:
2: $\quad \tilde{r}_u \leftarrow r_u$
3: $\quad x_u \leftarrow x_u + \tilde{r}_u$
4: $\quad r_u \leftarrow 0$
5: $\quad$ **for** $v \in \mathcal{N}(u)$ **do**
6: $\qquad r_v \leftarrow r_v + \frac{(1 - \alpha)\tilde{r}_u}{d_u}$
7: $\quad$ **Return** $(\boldsymbol{r}, \boldsymbol{x})$

---

We adopt the typical implementation of APPR as presented in Algo. 1.

---

[5] We will arbitrarily write $\alpha_{\text{PPR}}$ and $\alpha_{\text{Katz}}$ as $\alpha$ when the context is clear.

## B.3 Missing proofs of LocalSOR

**Theorem 3.2** (Properties of local diffusion process via LocalSOR). *Let $Q \triangleq I - \beta P$ where $P \geq \mathbf{0}_{n \times n}$ and $P_{uv} \neq 0$ if $(u, v) \in \mathcal{E}$; 0 otherwise. Define maximal value $P_{\max} = \max_{u \in \mathcal{V}} \|Pe_u\|_1$. Assume that $r^{(0)} \geq \mathbf{0}$ is nonnegative and $P_{\max}, \beta$ are such that $\beta P_{\max} < 1$, given the updates of (6), then the local diffusion process of $\phi\left(x^{(t)}, r^{(t)}, \mathcal{S}_t; s, \epsilon, \mathcal{G}, \mathcal{A}_\theta = (\text{LocalSOR}, \omega)\right)$ with $\omega \in (0, 1)$ has the following properties*

1. ***Nonnegativity.*** $r^{(t+t_i)} \geq \mathbf{0}$ *for all $t \geq 0$ and $t_i = (i-1)/|\mathcal{S}_t|$ with $i = 1, 2, \ldots, |\mathcal{S}_t|$.*

2. ***Monotonicity property.*** $\|r^{(0)}\|_1 \geq \cdots \|r^{(t+t_i)}\|_1 \geq \|r^{(t+t_{i+1})}\|_1 \cdots$.

*If the local diffusion process converges (i.e., $\mathcal{S}_T = \emptyset$), then $T$ is bounded by*

$$T \leq \frac{1}{\omega \overline{\gamma}_T (1 - \beta P_{\max})} \ln \frac{\|r^{(0)}\|_1}{\|r^{(T)}\|_1}, \text{ where } \overline{\gamma}_T \triangleq \frac{1}{T} \sum_{t=0}^{T-1} \left\{ \gamma_t \triangleq \frac{\sum_{i=1}^{|\mathcal{S}_t|} r_{u_i}^{(t+t_i)}}{\|r^{(t)}\|_1} \right\}.$$

*Proof.* At each step $t$ and $t_i = (i-1)/|\mathcal{S}_t|$ where $i = 1, 2, \ldots, |\mathcal{S}_t|$, recall LocalSOR for solving $Qx = s$ has the following updates

$$x^{(t+t_{i+1})} = x^{(t+t_i)} + \frac{\omega \cdot r_{u_i}^{(t+t_i)}}{q_{u_i u_i}} e_{u_i}, \quad r^{(t+t_{i+1})} = r^{(t+t_i)} - \frac{\omega \cdot r_{u_i}^{(t+t_i)}}{q_{u_i u_i}} Q \cdot e_{u_i}.$$

Since we define $Q$ as $Q = I - \beta P$, where $P_{uu} = 0$ for all $u \in \mathcal{V}$, and using $q_{u_i u_i} = 1$, we continue to have the updates of $r$ as follows

$$r^{(t+t_{i+1})} = r^{(t+t_i)} - \frac{\omega \cdot r_{u_i}^{(t+t_i)}}{q_{u_i u_i}} \cdot Q e_{u_i}$$

$$= r^{(t+t_i)} - \omega \cdot r_{u_i}^{(t+t_i)} e_{u_i} + \omega \beta r_{u_i}^{(t+t_i)} \cdot P e_{u_i}.$$

By using induction, we show the nonnegativity of $r^{(t)}$. First of all, $r^{(0)} \geq \mathbf{0}$ by our assumption. Let us assume $r^{(t+t_i)} \geq \mathbf{0}$, then the above equation gives $r^{(t+t_{i+1})} \geq \mathbf{0}$ since $\omega \leq 1$ and $P \geq \mathbf{0}_{n \times n}$ by our assumption. Since we assume $\omega \leq 1$ and $P \geq \mathbf{0}_{n \times n}$, the above equation can be written as $r^{(t+t_{i+1})} + \omega \cdot r_{u_i}^{(t+t_i)} e_{u_i} = r^{(t+t_i)} + \omega \beta r_{u_i}^{(t+t_i)} \cdot P e_{u_i}$. By taking $\| \cdot \|_1$ on both sides, we have

$$\|r^{(t+t_{i+1})}\|_1 + \omega r_{u_i}^{(t+t_i)} = \|r^{(t+t_i)}\|_1 + \omega \beta r_{u_i}^{(t+t_i)} \|P e_{u_i}\|_1$$

To make an effective reduction, $\beta$ should be such that $\beta \|P e_{u_i}\|_1 < 1$ for all $u_i \in \mathcal{V}$. Summing over $i = 1, 2, \ldots, |\mathcal{S}_t|$ of the above equation, we then have the following

$$\|r^{(t+1)}\|_1 = \left(1 - \frac{\omega \sum_{i=1}^{|\mathcal{S}_t|} r_{u_i}^{(t+t_i)} (1 - \beta \|P e_{u_i}\|_1)}{\|r^{(t)}\|_1}\right) \|r^{(t)}\|_1$$

$$\leq (1 - \omega \gamma_t (1 - \beta P_{\max})) \|r^{(t)}\|_1. \tag{17}$$

Note that we defined $\gamma_t = \frac{\sum_{i=1}^{|\mathcal{S}_t|} r_{u_i}^{(t+t_i)}}{\|r^{(t)}\|_1}$ and $(1 - \beta P\max) \leq (1 - \beta P_{u_i})$ for all $u_i \in \mathcal{S}_t$. The inequality of (17) leads to the following bound

$$\|r^{(T)}\|_1 \leq \prod_{t=0}^{T-1} (1 - \omega \gamma_t (1 - \beta P_{\max})) \|r^{(0)}\|_1.$$

To further simply the above upper bound, since each term $1 - \omega \gamma_t (1 - \beta P_{\max}) \geq 0$ during the updates, it follows that $\omega \gamma_t (1 - \beta P_{\max}) \in (0, 1)$. If there exists $T$ such that $\mathcal{S}_T = \emptyset$, then we can obtain an upper bound of $T$ as

$$\ln \frac{\|r^{(T)}\|_1}{\|r^{(0)}\|_1} \leq \sum_{t=0}^{T-1} \ln (1 - \omega \gamma_t (1 - \beta P_{\max}))$$

$$\leq - \sum_{t=0}^{T-1} \omega \gamma_t (1 - \beta P_{\max}),$$

which leads to

$$T \leq \frac{1}{\omega \overline{\gamma}_T (1 - \beta P_{\max})} \ln \frac{\|\boldsymbol{r}^{(0)}\|_1}{\|\boldsymbol{r}^{(T)}\|_1}.$$

$\square$

**Theorem 3.3** (Sublinear runtime bound of LocalSOR for PPR). *Let* $\mathcal{I}_T = \mathrm{supp}(\boldsymbol{r}^{(T)})$. *Given an undirected graph* $\mathcal{G}$ *and a target source node* $s$ *with* $\alpha \in (0,1), \omega = 1$, *and provided* $0 < \epsilon \leq 1/d_s$, *the run time of LocalSOR in Equ.* (6) *for solving* $(\boldsymbol{I} - (1-\alpha)\boldsymbol{A}\boldsymbol{D}^{-1})\boldsymbol{f}_{PPR} = \alpha \boldsymbol{e}_s$ *with the stop condition* $\|\boldsymbol{D}^{-1}\boldsymbol{r}^{(T)}\|_\infty \leq \alpha \epsilon$ *and initials* $\boldsymbol{x}^{(0)} = \boldsymbol{0}$ *and* $\boldsymbol{r}^{(0)} = \alpha \boldsymbol{e}_s$ *is bounded as the following*

$$\mathcal{T}_{LocalSOR} \leq \min \left\{ \frac{1}{\epsilon \alpha}, \frac{\overline{\mathrm{vol}}(\mathcal{S}_T)}{\alpha \overline{\gamma}_T} \ln \frac{C_{PPR}}{\epsilon} \right\}, \qquad \text{where } \frac{\overline{\mathrm{vol}}(\mathcal{S}_T)}{\overline{\gamma}_T} \leq \frac{1}{\epsilon}. \tag{18}$$

*where* $C_{PPR} = 1/((1-\alpha)|\mathcal{I}_T|)$. *The estimate* $\boldsymbol{x}^{(T)}$ *satisfies* $\|\boldsymbol{D}^{-1}(\boldsymbol{x}^{(T)} - \boldsymbol{f}_{PPR})\|_\infty \leq \epsilon$.

*Proof.* Recall $\mathcal{S}_t = \{u_1, u_2, \ldots, u_{|\mathcal{S}_t|}\}$ be the set of active nodes processed in $t$ iteration. For convenience, we denote $|\mathcal{S}_t| = k$. By LocalSOR defined in (6) for solving

$$\underbrace{\left(\boldsymbol{I} - (1-\alpha)\boldsymbol{A}\boldsymbol{D}^{-1}\right)}_{\boldsymbol{Q}} \boldsymbol{f}_{\mathrm{PPR}} = \underbrace{\alpha \boldsymbol{e}_s}_{\boldsymbol{s}}.$$

We have the following online iteration updates for each active node $u_i \in \mathcal{S}_t$ (recall $q_{u_i u_i} = 1$).

$$\boldsymbol{x}^{(t+\frac{i}{k})} = \boldsymbol{x}^{(t+\frac{i-1}{k})} + \omega \cdot r_{u_i}^{(t+\frac{i-1}{k})} \cdot \boldsymbol{e}_{u_i}, \quad \text{for } i = 1, \ldots, k \text{ and } u_i \in \mathcal{S}_t,$$

$$\boldsymbol{r}^{(t+\frac{i}{k})} = \boldsymbol{r}^{(t+\frac{i-1}{k})} - \omega \cdot r_{u_i}^{(t+\frac{i-1}{k})} \cdot \boldsymbol{e}_{u_i} + \omega \cdot (1-\alpha) \cdot r_{u_i}^{(t+\frac{i-1}{k})} \cdot \boldsymbol{A}\boldsymbol{D}^{-1}\boldsymbol{e}_{u_i}.$$

Note for each active node $u_i$, we have $r_{u_i}^{(t+\frac{i-1}{k})} \geq \epsilon \alpha d_{u_i}$. The total operations for LocSOR is

$$\begin{aligned}
\mathcal{T}_{\mathrm{LocalSOR}} &:= \sum_{t=0}^{T-1} \mathrm{vol}(\mathcal{S}_t) \\
&= \sum_{t=0}^{T-1} \sum_{i=1}^{k} d_{u_i} \\
&\leq \sum_{t=0}^{T-1} \sum_{i=1}^{k} \frac{r_{u_i}^{(t+(i-1)/k))}}{\epsilon \alpha} \\
&= \sum_{t=0}^{T-1} \sum_{i=1}^{k} \frac{\|\boldsymbol{r}^{(t+\frac{i-1}{k})}\|_1 - \|\boldsymbol{r}^{(t+\frac{i}{k})}\|_1}{\omega \epsilon \alpha^2} \\
&= \frac{\|\boldsymbol{r}^{(0)}\|_1 - \|\boldsymbol{r}^{(T)}\|_1}{\omega \epsilon \alpha^2} \\
&\leq \frac{\|\boldsymbol{r}^{(0)}\|_1}{\omega \epsilon \alpha^2} \\
&= \frac{1}{\epsilon \alpha},
\end{aligned}$$

where the last equality follows from $\omega = 1$ and $\|\boldsymbol{r}^{(0)}\|_1 = \alpha$. Next, we focus on the proof of our newly derived bound $\tilde{O}(\overline{\mathrm{vol}}(\mathcal{S}_T)/(\alpha \overline{\gamma} T))$. To check the upper bound of $T$, from Theorem 3.2 by noticing $\beta = (1-\alpha)$ and $P_{\max} = 1$, this leads to the following

$$\begin{aligned}
T &\leq \frac{1}{\omega \overline{\gamma}_T (1 - \beta P_{\max})} \ln \frac{\|\boldsymbol{r}^{(0)}\|_1}{\|\boldsymbol{r}^{(T)}\|_1} \\
&= \frac{1}{\alpha \overline{\gamma}_T} \ln \frac{\|\boldsymbol{r}^{(0)}\|_1}{\|\boldsymbol{r}^{(T)}\|_1}.
\end{aligned}$$

Next, we try to give a lower bound of $\boldsymbol{r}^{(T)}$. Note that after the last iteration $T$, for each nonzero residual $r_u^{(T)} \neq 0, u \in \mathcal{I}_T$, there is at least one possible update that happened at node $u$: 1) Node $u$ has a neighbor $v_u \in \mathcal{N}(u)$, which was active. This neighbor $v_u$ pushed some residual $(1-\alpha)r_{v_u}^{(t')}/d_{v_u}$ to $u$ where the time $t' < T$. Hence, for all $u \in \mathcal{I}_T$, we have

$$
\begin{aligned}
\|\boldsymbol{r}^{(T)}\|_1 &= \sum_{u \in \mathcal{I}_T} r_u^{(T)} \\
&\geq \sum_{u \in \mathcal{I}_T} \frac{(1-\alpha)r_{v_u}^{(t')}}{d_{v_u}} \\
&\geq \sum_{u \in \mathcal{I}_T} \frac{(1-\alpha)\epsilon\alpha d_{v_u}}{d_{v_u}} \\
&= \sum_{u \in \mathcal{I}_T} (1-\alpha)\epsilon\alpha \\
&\geq \alpha\epsilon(1-\alpha)|\mathcal{I}_T|,
\end{aligned}
$$

where the first equality is due to the nonnegativity of $\boldsymbol{r}$ guaranteeing by Theorem 3.2 and the second inequality is due to the fact that $r_u^{(t')}$ was active residuals before the push operation. Applying the above lower bound of $\|\boldsymbol{r}^{(T)}\|_1$, we obtain

$$
\begin{aligned}
\frac{\|\boldsymbol{r}^{(0)}\|_1}{\|\boldsymbol{r}^{(T)}\|_1} &\leq \frac{\|\boldsymbol{r}^{(0)}\|_1}{\alpha\epsilon(1-\alpha)|\mathcal{I}_T|} \\
&= \frac{1}{\epsilon(1-\alpha)|\mathcal{I}_T|} \\
&:= \frac{C_{\mathrm{PPR}}}{\epsilon},
\end{aligned}
$$

where $C_{\mathrm{PPR}} = 1/((1-\alpha)|\mathcal{I}_T|)$ and $\mathcal{I}_T = \mathrm{supp}(\boldsymbol{r}^{(T)})$. Combine the above inequality and the upper bound $T$, we obtain our new local diffusion-based bounds. The rest is to show a lower bound of $1/\epsilon$. By using the active node condition, for $u_i \in \mathcal{S}_t$, we have

$$
\alpha\epsilon\,\mathrm{vol}(\mathcal{S}_t) \leq \sum_{i=1}^{|\mathcal{S}_t|} r_{u_i}^{(t+t_i)}
$$

$$
\implies \alpha\epsilon \sum_{t=0}^{T-1} \mathrm{vol}(\mathcal{S}_t) \leq \sum_{t=0}^{T-1} \sum_{i=1}^{|\mathcal{S}_t|} r_{u_i}^{(t+t_i)}. \tag{19}
$$

We continue to have

$$
\begin{aligned}
\frac{\overline{\mathrm{vol}}(\mathcal{S}_T)}{\overline{\gamma}_T} &= \frac{\sum_{t=0}^{T-1} \mathrm{vol}(\mathcal{S}_t)}{\sum_{t=0}^{T-1} \gamma_t} \\
&\leq \frac{\sum_{t=0}^{T-1} \mathrm{vol}(\mathcal{S}_t)}{\sum_{t=0}^{T-1} \frac{\sum_{i=1}^{|\mathcal{S}_t|} r_{u_i}^{(t+t_i)}}{\|\boldsymbol{r}^{(0)}\|_1}} \\
&= \alpha \frac{\sum_{t=0}^{T-1} \mathrm{vol}(\mathcal{S}_t)}{\sum_{t=0}^{T-1} \sum_{i=1}^{|\mathcal{S}_t|} r_{u_i}^{(t+t_i)}} \\
&\leq \frac{1}{\epsilon},
\end{aligned}
$$

where the first inequality follows from the fact that $\frac{\sum_{i=1}^{|\mathcal{S}_t|} r_{u_i}^{(t+t_i)}}{\|\boldsymbol{r}^{(0)}\|_1} \leq \frac{\sum_{i=1}^{|\mathcal{S}_t|} r_{u_i}^{(t+t_i)}}{\|\boldsymbol{r}^{(t)}\|_1}$ for $t = 0, 1, \ldots, T-1$, due to the monotonicity property of $\|\boldsymbol{r}^{(t)}\|_1$. The last inequality is from (19). Combining these, we finish the proof of the sublinear runtime bound. The rest is to prove the estimation quality. Note $\|\boldsymbol{D}^{-1}\boldsymbol{r}^{(T)}\|_\infty \leq \alpha\epsilon$ directly implies $\|\boldsymbol{D}^{-1}(\boldsymbol{x}^{(T)} - \boldsymbol{f}_{\mathrm{PPR}})\|_\infty \leq \epsilon$ by using Proposition B.1 and the corresponding stop condition. $\qquad\square$

*Remark* B.2. The above theorem shares the proof strategy we provided in [77]. Here, we use a slightly different formulation of the linear system.

**Corollary 3.4** (Runtime bound of LocalSOR for Katz). *Let $\mathcal{I}_T = \mathrm{supp}(\boldsymbol{r}^{(T)})$ and $C_{Katz} = 1/((1 - \alpha)|\mathcal{I}_T|)$. Given an undirected graph $\mathcal{G}$ and a target source node $s$ with $\alpha \in (0, 1/d_{\max}), \omega = 1$, and provided $0 < \epsilon \leq 1/d_s$, the run time of LocalSOR in Equ. (6) for solving $(\boldsymbol{I} - \alpha\boldsymbol{A})\boldsymbol{x} = \boldsymbol{e}_s$ with the stop condition $\|\boldsymbol{D}^{-1}\boldsymbol{r}^{(T)}\|_\infty \leq \epsilon$ and initials $\boldsymbol{x}^{(0)} = \boldsymbol{0}$ and $\boldsymbol{r}^{(0)} = \boldsymbol{e}_s$ is bounded as the following*

$$\mathcal{T}_{LocalSOR} \leq \min\left\{\frac{1}{\epsilon(1 - \alpha d_{\max})}, \frac{\overline{\mathrm{vol}}(\mathcal{S}_T)}{(1 - \alpha d_{\max})\overline{\gamma}_T} \ln\frac{C_{Katz}}{\epsilon}\right\}, \ \textit{where} \ \frac{\overline{\mathrm{vol}}(\mathcal{S}_T)}{\overline{\gamma}_T} \leq \frac{1}{\epsilon}. \qquad (20)$$

*The estimate $\hat{\boldsymbol{f}}_{Katz} = \boldsymbol{x}^{(T)} - \boldsymbol{e}_s$ satisfies $\|\hat{\boldsymbol{f}}_{Katz} - \boldsymbol{f}_{Katz}\|_2 \leq \|(\boldsymbol{I} - \alpha\boldsymbol{A})^{-1}\boldsymbol{D}\|_1\epsilon$.*

*Proof.* The linear system $(\boldsymbol{I} - \alpha\boldsymbol{A})\boldsymbol{x} = \boldsymbol{s}$ using LocalSOR updates can be written as

$$\boldsymbol{x}^{(t+t_i+\Delta t)} = \boldsymbol{x}^{(t+t_i)} + \frac{\omega r_{u_i}^{(t+t_i)}}{q_{u_i u_i}}\boldsymbol{e}_{u_i},$$

$$\boldsymbol{r}^{(t+t_i+\Delta t)} = \boldsymbol{r}^{(t+t_i)} - \frac{\omega r_{u_i}^{(t+t_i)}}{q_{u_i u_i}}\boldsymbol{e}_{u_i} + \omega r_{u_i}^{(t+t_i)}\alpha\boldsymbol{A}\frac{\boldsymbol{e}_{u_i}}{q_{u_i u_i}},$$

where $q_{u_i u_i} = 1$. The updates of $\boldsymbol{r}^{(t+t_i)}$ can be simplified as the following

$$\boldsymbol{r}^{(t+t_i+\Delta t)} = \boldsymbol{r}^{(t+t_i)} - \omega r_{u_i}^{(t+t_i)}\boldsymbol{e}_{u_i} + \omega r_{u_i}^{(t+t_i)}\alpha\boldsymbol{A}\boldsymbol{e}_{u_i}.$$

The proof of nonnegativity of $\boldsymbol{r}^{(t)}$ follows the similar induction as in previous theorem by noticing the spectral radius of $\boldsymbol{A}$ is $|\lambda(\boldsymbol{A})| \leq d_{\max}$. Hence, we have

$$\boldsymbol{r}^{(t+1)} = \boldsymbol{r}^{(t)} - \omega\sum_{i=1}^{|\mathcal{S}_t|} r_{u_i}^{(t+t_i)}\boldsymbol{e}_{u_i} + \omega\sum_{i=1}^{|\mathcal{S}_t|} r_{u_i}^{(t+t_i)}\alpha\boldsymbol{A}\boldsymbol{e}_{u_i},$$

Move the negative term to left and take $\ell_1$-norm on both sides, we have

$$\omega\sum_{i=1}^{|\mathcal{S}_t|} r_{u_i}^{(t+t_i)} = \|\boldsymbol{r}^{(t)}\|_1 - \|\boldsymbol{r}^{(t+1)}\|_1 + \omega\sum_{i=1}^{|\mathcal{S}_t|} r_{u_i}^{(t+t_i)}\alpha d_{u_i}$$

$$\|\boldsymbol{r}^{(t+1)}\|_1 = \left(1 - \frac{\omega\sum_{i=1}^{|\mathcal{S}_t|} r_{u_i}^{(t+t_i)} - \omega\sum_{i=1}^{|\mathcal{S}_t|} r_{u_i}^{(t+t_i)}\alpha d_{u_i}}{\|\boldsymbol{r}^{(t)}\|_1}\right)\|\boldsymbol{r}^{(t)}\|_1$$

$$\leq \left(1 - \frac{\omega(1 - \alpha d_{\max})\sum_{i=1}^{|\mathcal{S}_t|} r_{u_i}^{(t+t_i)}}{\|\boldsymbol{r}^{(t)}\|_1}\right)\|\boldsymbol{r}^{(t)}\|_1.$$

It leads to the following

$$\omega\sum_{i=1}^{|\mathcal{S}_t|} r_{u_i}^{(t+t_i)} - \omega\sum_{i=1}^{|\mathcal{S}_t|} r_{u_i}^{(t+t_i)}\alpha d_{u_i} = \|\boldsymbol{r}^{(t)}\|_1 - \|\boldsymbol{r}^{(t+1)}\|_1$$

Since each $r_{u_i}^{(t+t_i)} \geq \epsilon d_{u_i}$, $\mathrm{vol}(\mathcal{S}_t) \leq \sum_{i=1}^{|\mathcal{S}_t|} r_{u_i}^{(t+t_i)}/\epsilon$, we continue have

$$\omega\sum_{i=1}^{|\mathcal{S}_t|} r_{u_i}^{(t+t_i)} - \omega\sum_{i=1}^{|\mathcal{S}_t|} r_{u_i}^{(t+t_i)}\alpha d_{u_i} = \omega\sum_{i=1}^{|\mathcal{S}_t|} r_{u_i}^{(t+t_i)}\left(1 - \alpha d_{u_i}\right)$$

$$\geq \omega\sum_{i=1}^{|\mathcal{S}_t|} r_{u_i}^{(t+t_i)}\left(1 - \alpha d_{\max}\right).$$

Finally, the above inequality gives the following upper runtime bound as

$$\sum_{t=0}^{T-1} \mathrm{vol}(\mathcal{S}_t) \leq \sum_{t=0}^{T-1} \frac{\|\boldsymbol{r}^{(t)}\|_1 - \|\boldsymbol{r}^{(t+1)}\|_1}{\omega\epsilon(1 - \alpha d_{\max})}$$

$$= \frac{\|\boldsymbol{r}^{(0)}\|_1 - \|\boldsymbol{r}^{(T)}\|_1}{\omega\epsilon(1 - \alpha d_{\max})}$$

$$\leq \frac{1}{\omega\epsilon(1 - \alpha d_{\max})}.$$

By letting $\beta = \alpha$ and $\boldsymbol{P} = \boldsymbol{A}$ with $P_{\max} = d_{\max}$, we apply Theorem 3.2, to obtain the local diffusion-based bound as

$$\|\boldsymbol{r}^{(t+1)}\|_1 \leq (1 - \omega(1 - \alpha d_{\max})\beta_t)\,\|\boldsymbol{r}^{(t)}\|_1$$

$$\|\boldsymbol{r}^{(T)}\|_1 \leq \prod_{t=0}^{T-1} (1 - \omega(1 - \alpha d_{\max})\beta_t)\,\|\boldsymbol{r}^{(0)}\|_1$$

$$= \prod_{t=0}^{T-1} (1 - \omega(1 - \alpha d_{\max})\beta_t)\,.$$

Using the similar technique provided in the previous case, and letting $\omega = 1$ we can have

$$\mathcal{T}_{\mathrm{Katz}} \leq \frac{\overline{\mathrm{vol}}(\mathcal{S}_T)}{(1 - \alpha d_{\max})\overline{\gamma}_T} \ln \frac{C_{\mathrm{Katz}}}{\epsilon}\,.$$

To check the estimate equality, we have

$$\boldsymbol{r}^{(T)} = \boldsymbol{e}_s - (\boldsymbol{I} - \alpha\boldsymbol{A})\boldsymbol{x}^{(T)}$$

$$\implies (\boldsymbol{I} - \alpha\boldsymbol{A})^{-1}\boldsymbol{r}^{(T)} = \left((\boldsymbol{I} - \alpha\boldsymbol{A})^{-1} - \boldsymbol{I}\right)\boldsymbol{e}_s + \boldsymbol{e}_s - \boldsymbol{x}^{(T)}$$

$$= \boldsymbol{f}_{\mathrm{Katz}} + \boldsymbol{e}_s - \boldsymbol{x}^{(T)}\,.$$

Let $\hat{\boldsymbol{f}}_{\mathrm{Katz}} := \boldsymbol{x}^{(T)} - \boldsymbol{e}_s$ This leads to

$$\|\hat{\boldsymbol{f}}_{\mathrm{Katz}} - \boldsymbol{f}_{\mathrm{Katz}}\|_2 = \|(\boldsymbol{I} - \alpha\boldsymbol{A})^{-1}\boldsymbol{D}\boldsymbol{D}^{-1}\boldsymbol{r}^{(T)}\|_2$$

$$\leq \|(\boldsymbol{I} - \alpha\boldsymbol{A})^{-1}\boldsymbol{D}\|_1 \cdot \|\boldsymbol{D}^{-1}\boldsymbol{r}^{(T)}\|_\infty$$

$$\leq \|(\boldsymbol{I} - \alpha\boldsymbol{A})^{-1}\boldsymbol{D}\|_1\epsilon\,.$$

$\square$

## B.4 Missing proofs of LocalGD

**Theorem 3.5** (Properties of local diffusion process via LocalGD)**.** *Let $\boldsymbol{Q} \triangleq \boldsymbol{I} - \beta\boldsymbol{P}$ where $\boldsymbol{P} \geq \boldsymbol{0}_{n\times n}$ and $P_{uv} \neq 0$ if $(u,v) \in \mathcal{E}$; 0 otherwise. Define maximal value $P_{\max} = \max_{\mathcal{S}_t \subseteq \mathcal{V}} \|\boldsymbol{P}\boldsymbol{r}_{\mathcal{S}_t}\|_1/\|\boldsymbol{r}_{\mathcal{S}_t}\|_1$. Assume that $\boldsymbol{r}^{(0)} = \boldsymbol{s} \geq \boldsymbol{0}$ and $P_{\max}, \beta$ are such that $\beta P_{\max} < 1$, given the updates of (10), then the local diffusion process of $\phi\left(\mathcal{S}_t, \boldsymbol{x}^{(t)}, \boldsymbol{r}^{(t)}; \mathcal{G}, \mathcal{A}_\theta = (\mathrm{LocalGD}, \mu, L)\right)$ has the following properties*

1. **Nonnegativity.** *$\boldsymbol{r}^{(t)} \geq \boldsymbol{0}$ for all $t \geq 0$.*

2. **Monotonicity property.** *$\|\boldsymbol{r}^{(0)}\|_1 \geq \cdots \|\boldsymbol{r}^{(t)}\|_1 \geq \|\boldsymbol{r}^{(t+1)}\|_1 \cdots$ .*

*If the local diffusion process converges (i.e., $\mathcal{S}_T = \emptyset$), then $T$ is bounded by*

$$T \leq \frac{1}{\overline{\gamma}_T(1 - \beta P_{\max})} \ln \frac{\|\boldsymbol{r}^{(0)}\|_1}{\|\boldsymbol{r}^{(T)}\|_1}, \quad \text{where } \overline{\gamma}_T \triangleq \frac{1}{T}\sum_{t=0}^{T-1}\left\{\gamma_t \triangleq \frac{\|\boldsymbol{r}^{(t)}_{\mathcal{S}_t}\|_1}{\|\boldsymbol{r}^{(t)}\|_1}\right\}\,.$$

*Proof.* At each step $t$, recall LocalGD for solving $\boldsymbol{Q}\boldsymbol{x} = (\boldsymbol{I} - \beta\boldsymbol{P})\boldsymbol{x} = \boldsymbol{s}$ has the following updates

$$\boldsymbol{x}^{(t+1)} = \boldsymbol{x}^{(t)} + \boldsymbol{r}^{(t)}_{\mathcal{S}_t}, \qquad \boldsymbol{r}^{(t+1)} = \boldsymbol{r}^{(t)} - \boldsymbol{Q}\boldsymbol{r}^{(t)}_{\mathcal{S}_t}\,.$$

Since $\boldsymbol{Q} = \boldsymbol{I} - \beta\boldsymbol{P}$, we continue to have the updates of $\boldsymbol{r}$ as the following

$$\boldsymbol{r}^{(t+1)} = \boldsymbol{r}^{(t)} - (\boldsymbol{I} - \beta\boldsymbol{P})\boldsymbol{r}^{(t)}_{\mathcal{S}_t}$$

$$= \boldsymbol{r}^{(t)} - \boldsymbol{r}^{(t)}_{\mathcal{S}_t} + \beta\boldsymbol{P}\boldsymbol{r}^{(t)}_{\mathcal{S}_t}\,.$$

By using induction, we show the nonnegativity of $\boldsymbol{r}^{(t)}$. First of all, $\boldsymbol{r}^{(0)} \geq \boldsymbol{0}$ by our assumption. Let us assume $\boldsymbol{r}^{(t)} \geq \boldsymbol{0}$. Then, the above equation gives $\boldsymbol{r}^{(t+1)} \geq \boldsymbol{0}$ since $\beta \geq 0$ and $\boldsymbol{P} \geq \boldsymbol{0}_{n\times n}$ by our assumption. The above equation can be written as

$$\boldsymbol{r}^{(t+1)} + \boldsymbol{r}^{(t)}_{\mathcal{S}_t} = \boldsymbol{r}^{(t)} + \beta\boldsymbol{P}\boldsymbol{r}^{(t)}_{\mathcal{S}_t}$$

Taking $\|\cdot\|_1$ on both sides, we have

$$\|\boldsymbol{r}^{(t+1)}\|_1 + \|\boldsymbol{r}_{\mathcal{S}_t}^{(t)}\|_1 = \|\boldsymbol{r}^{(t)}\|_1 + \beta\|\boldsymbol{Pr}_{\mathcal{S}_t}^{(t)}\|_1.$$

To make an effective reduction, $\beta$ should be such that $\beta\|\boldsymbol{Pr}_{\mathcal{S}_t}^{(t)}\|_1 < \|\boldsymbol{r}_{\mathcal{S}_t}^{(t)}\|_1$ for all $\mathcal{S}_t \subseteq \mathcal{V}$. For this, we then have the following

$$\|\boldsymbol{r}^{(t+1)}\|_1 = \left(1 - \frac{\|\boldsymbol{r}_{\mathcal{S}_t}^{(t)}\|_1 - \beta\|\boldsymbol{Pr}_{\mathcal{S}_t}^{(t)}\|_1}{\|\boldsymbol{r}^{(t)}\|_1}\right)\|\boldsymbol{r}^{(t)}\|_1$$

$$\leq (1 - \gamma_t(1 - \beta P_{\max}))\|\boldsymbol{r}^{(t)}\|_1, \tag{21}$$

where note we defined $\gamma_t = \frac{\|\boldsymbol{r}_{\mathcal{S}_t}^{(t)}\|_1}{\|\boldsymbol{r}^{(t)}\|_1}$ and assumed $\|\boldsymbol{Pr}_{\mathcal{S}_t}^{(t)}\|_1 \leq P_{\max}\|\boldsymbol{r}_{\mathcal{S}_t}^{(t)}\|_1$. Apply (21) from $t = 0$ to $T$, it leads to the following bound

$$\|\boldsymbol{r}^{(T)}\|_1 \leq \prod_{t=0}^{T-1}(1 - \gamma_t(1 - \beta P_{\max}))\|\boldsymbol{r}^{(0)}\|_1$$

Note each of the term $1 - \gamma_t(1 - \beta P_{\max}) \geq 0$ during the updates, then $\gamma_t(1 - \beta P_{\max}) \in (0, 1)$. To check the upper bound of $T$, we have

$$\ln\frac{\|\boldsymbol{r}^{(T)}\|_1}{\|\boldsymbol{r}^{(0)}\|_1} \leq \sum_{t=0}^{T-1}\ln(1 - \gamma_t(1 - \beta P_{\max}))$$

$$\leq -\sum_{t=0}^{T-1}\gamma_t(1 - \beta P_{\max}),$$

which leads to

$$T \leq \frac{1}{\overline{\gamma}_T(1 - \beta P_{\max})}\ln\frac{\|\boldsymbol{r}^{(0)}\|_1}{\|\boldsymbol{r}^{(T)}\|_1}.$$

$\square$

**Corollary 3.6** (Convergence of LocalGD for PPR and Katz). *Let $\mathcal{I}_T = \mathrm{supp}(\boldsymbol{r}^{(T)})$ and $C = \frac{1}{(1-\alpha)|\mathcal{I}_T|}$. Use LocalGD to approximate PPR or Katz by using iterative procedure* (10). *Denote $\mathcal{T}_{PPR}$ and $\mathcal{T}_{Katz}$ as the total number of operations needed by using LocalGD, they can then be bounded by*

$$\mathcal{T}_{PPR} \leq \min\left\{\frac{1}{\alpha_{PPR}\cdot\epsilon}, \frac{\overline{\mathrm{vol}}(\mathcal{S}_T)}{\alpha_{PPR}\cdot\overline{\gamma}_T}\ln\frac{C}{\epsilon}\right\}, \quad \frac{\overline{\mathrm{vol}}(\mathcal{S}_T)}{\overline{\gamma}_T} \leq \frac{1}{\epsilon} \tag{22}$$

*for a stop condition $\|\boldsymbol{D}^{-1}\boldsymbol{r}^{(t)}\|_\infty \leq \alpha_{PPR}\cdot\epsilon$. For solving* KATZ, *then the toal runtime is bounded by*

$$\mathcal{T}_{Katz} \leq \min\left\{\frac{1}{(1 - \alpha_{Katz}\cdot d_{\max})\epsilon}, \frac{\overline{\mathrm{vol}}(\mathcal{S}_T)}{(1 - \alpha_{Katz}\cdot d_{\max})\overline{\gamma}_T}\ln\frac{C_2}{\epsilon}\right\}, \quad \frac{\overline{\mathrm{vol}}(\mathcal{S}_T)}{\overline{\gamma}_T} \leq \frac{1}{\epsilon} \tag{23}$$

*for a stop condition $\|\boldsymbol{D}^{-1}\boldsymbol{r}^{(t)}\|_\infty \leq \epsilon d_u$. The estimate equality is the same as of LocalSOR.*

*Proof.* We first show graph-independent bound $\mathcal{O}(1/(\alpha\epsilon))$ for PPR computation. Since $\boldsymbol{r}^{(t+1)} = \boldsymbol{r}^{(t)} - \boldsymbol{Qr}_{\mathcal{S}_t}^{(t)}$, then rearrange it to[6]

$$\boldsymbol{r}^{(t+1)} + \boldsymbol{r}_{\mathcal{S}_t}^{(t)} = \boldsymbol{r}^{(t)} + (1 - \alpha)\boldsymbol{AD}^{-1}\boldsymbol{r}_{\mathcal{S}_t}^{(t)}.$$

Note that entries in $\boldsymbol{r}^{(t)}$ are nonnegative. It leads to

$$\|\boldsymbol{r}^{(t+1)}\|_1 + \|\boldsymbol{r}_{\mathcal{S}_t}^{(t)}\|_1 = \|\boldsymbol{r}^{(t)}\|_1 + \|(1 - \alpha)\boldsymbol{AD}^{-1}\boldsymbol{r}_{\mathcal{S}_t}^{(t)}\|_1,$$

---

[6]To simplify the proof, here $\boldsymbol{r}$ represents $\boldsymbol{D}^{1/2}\boldsymbol{r}$.

where note $\|(1-\alpha)\boldsymbol{A}\boldsymbol{D}^{-1}\boldsymbol{r}_{\mathcal{S}_t}^{(t)}\|_1 = (1-\alpha)\|\boldsymbol{r}_{\mathcal{S}_t}^{(t)}\|_1$, then we have

$$\|\boldsymbol{r}_{\mathcal{S}_t}^{(t)}\|_1 = (\|\boldsymbol{r}^{(t)}\|_1 - \|\boldsymbol{r}^{(t+1)}\|_1)/\alpha.$$

At step $t$, LocalGD accesses the indices in $\mathcal{S}_t$. By the active node condition $\epsilon\alpha d_{u_i} \leq r_{u_i}^{(t)}$, we have

$$
\begin{aligned}
\text{vol}(\mathcal{S}_t) &= \sum_{i=1}^{k} d_{u_i} \\
&\leq \sum_{i=1}^{k} \frac{r_{u_i}^{(t)}}{\epsilon\alpha} \\
&= \frac{\|\boldsymbol{r}_{\mathcal{S}_t}^{(t)}\|_1}{\epsilon\alpha} \\
&= \frac{\|\boldsymbol{r}^{(t)}\|_1 - \|\boldsymbol{r}^{(t+1)}\|_1}{\epsilon\alpha^2}.
\end{aligned}
$$

Then the total run time of LocalGD is

$$
\begin{aligned}
\mathcal{T}_{\text{PPR}} &:= \sum_{t=0}^{T-1} \text{vol}(\mathcal{S}_t) \\
&\leq \frac{1}{\epsilon\alpha^2} \sum_{t=0}^{T-1} \left( \|\boldsymbol{r}^{(t)}\|_1 - \|\boldsymbol{r}^{(t+1)}\|_1 \right) \\
&= \frac{\|\boldsymbol{r}^{(0)}\|_1 - \|\boldsymbol{r}^{(T)}\|_1}{\epsilon\alpha^2} \\
&\leq \frac{1}{\epsilon\alpha},
\end{aligned}
$$

where the last inequality is due to $\|\boldsymbol{r}^{(t)}\|_1 = \alpha$. Therefore, the total run time is $\mathcal{O}(1/(\alpha\epsilon))$. Follow the similar technique, we have sublinear runtime bound for Katz, i.e., $\mathcal{T}_{\text{Kata}} \leq \frac{1}{(1-\alpha_{\text{Katz}}\cdot d_{\max})\epsilon}$. Next, we show the local diffusion bound for PPR. Recall LocalGD has the initial $\boldsymbol{x}^{(0)} = \boldsymbol{0}, \boldsymbol{r}^{(0)} = \alpha\boldsymbol{e}_s$ and the following updates

$$\boldsymbol{x}^{(t+1)} = \boldsymbol{x}^{(t)} + \boldsymbol{r}_{\mathcal{S}_t}^{(t)}, \quad \boldsymbol{r}^{(t+1)} = \boldsymbol{r}^{(t)} - \boldsymbol{Q}\boldsymbol{r}_{\mathcal{S}_t}^{(t)}, \quad \mathcal{S}_t = \left\{ u : r_u^{(t)} \geq \epsilon\alpha d_u, u \in \mathcal{V} \right\}.$$

Since $\beta = 1 - \alpha$ and $\boldsymbol{P} = \boldsymbol{A}\boldsymbol{D}^{-1}$, by applying Theorem 3.5, we have the upper bound of $T$

$$\|\boldsymbol{r}^{(T)}\|_1 = \prod_{t=0}^{T-1} (1-\alpha\gamma_t)\|\boldsymbol{r}^{(0)}\|_1 \qquad \Rightarrow \qquad T \leq \frac{1}{\alpha\overline{\gamma}_T} \ln \frac{\|\boldsymbol{r}^{(0)}\|_1}{\|\boldsymbol{r}^{(T)}\|_1}.$$

Note that each nonzero $r_u^{(T)}$ has at least part of the magnitude from the push operation of an active node. This means each nonzero of $\boldsymbol{r}^{(T)}$ satisfies

$$
\begin{aligned}
r_u^{(T)} &\geq \frac{(1-\alpha)r_v^{(\tilde{t})}}{d_v} \\
&\geq \frac{(1-\alpha)\alpha\epsilon d_v}{d_v} = (1-\alpha)\alpha\epsilon, \text{ for } u \in \mathcal{I}_T \\
\Rightarrow \quad \|\boldsymbol{r}^{(T)}\|_1 &\geq (1-\alpha)\alpha\epsilon|\mathcal{I}_T|,
\end{aligned}
$$

where $\tilde{t} \leq T$. Hence, we have

$$
\begin{aligned}
T &\leq \frac{1}{\alpha\overline{\gamma}_T} \ln \frac{\|\boldsymbol{r}^{(0)}\|_1}{\|\boldsymbol{r}^{(T)}\|_1} \\
&\leq \frac{1}{\alpha\overline{\gamma}_T} \ln \frac{\|\boldsymbol{r}^{(0)}\|_1}{\alpha\epsilon(1-\alpha)|\mathcal{I}_T|} \\
&:= \frac{1}{\alpha\overline{\gamma}_T} \ln \frac{C}{\epsilon},
\end{aligned}
$$

where $C = \frac{1}{(1-\alpha)|\mathcal{I}_T|}$. To see the additive error, note that $\boldsymbol{r}^{(T)} := \boldsymbol{b} - \boldsymbol{Q}\boldsymbol{x}^{(T)} = \alpha\boldsymbol{e}_s - \left(\boldsymbol{I} - (1-\alpha)\boldsymbol{A}\boldsymbol{D}^{-1}\right)\boldsymbol{x}^{(T)}$. It has the following equality

$$\boldsymbol{f}_s - \boldsymbol{x}^{(T)} = \left(\boldsymbol{I} - (1-\alpha)\boldsymbol{A}\boldsymbol{D}^{-1}\right)^{-1}\boldsymbol{r}^{(T)}$$

To estimate the bound $|f_s[u] - x_u^{(T)}|$, we need to know $\left(\left(\boldsymbol{I} - (1-\alpha)\boldsymbol{A}\boldsymbol{D}^{-1}\right)^{-1}\boldsymbol{r}^{(T)}\right)_u$. Specifically, the $u$-th entry of the above is

$$f_s[u] - x_u^{(T)} = \frac{1}{\alpha}\sum_{v \in \mathcal{V}} f_v[u]r_v^{(T)}$$

By Proposition B.1, we know $d_u f_u[v] = d_v f_v[u]$, then we continue to have

$$f_s[u] - x_u^{(T)} = \frac{1}{\alpha}\sum_{v \in \mathcal{V}} \frac{d_u f_u[v]}{d_v}r_v^{(T)}$$
$$\leq \frac{1}{\alpha}\sum_{v \in \mathcal{V}} d_u f_u[v]\epsilon\alpha$$
$$= \epsilon d_u \sum_{v \in \mathcal{V}} f_u[v]$$
$$= \epsilon d_u,$$

where the first inequality is due to $r_v^{(T)} < \epsilon\alpha d_v$, and the last equality follows from $\sum_{v \in \mathcal{V}} f_u[v] = 1$. Combining all inequalities for $u \in \mathcal{V}$, we have the estimate $|\boldsymbol{D}^{-1}(\hat{\boldsymbol{f}} - \boldsymbol{f}_{\mathrm{PPR}})|_1 \leq \epsilon$. We omit the proof of the sublinear runtime bound of LocalGD for Katz, as it largely follows the proof technique in Corollary 3.4. For the two lower bounds of $1/\epsilon$, i.e., $\overline{\mathrm{vol}}(\mathcal{S}_T)/\overline{\gamma}_T \leq \frac{1}{\epsilon}$, it directly follows from the monotonicity and nonnegativity properties stated in Theorem 3.2 and Theorem 3.5. $\square$

*Remark* B.3. The part of the theorem shares the similar proof strategy we provided in [77]. Here, we use a slightly different formulation of the linear system.

### B.5 Implementation Details

- **Chebyshev for PPR.** Let $\boldsymbol{Q} = \boldsymbol{I} - (1-\alpha)\boldsymbol{D}^{-1/2}\boldsymbol{A}\boldsymbol{D}^{-1/2}$ and $\boldsymbol{b} = \alpha\boldsymbol{D}^{-1/2}\boldsymbol{e}_s$. Then $\boldsymbol{f}^{(t)} = \boldsymbol{D}^{1/2}\boldsymbol{x}^{(t)}$. The eigenvalue of $\boldsymbol{Q}$ is in range $[\alpha, 2-\alpha]$. So, we let $L = 2 - \alpha$ and $\mu = \alpha$.

  1. $\delta_1 = 1 - \alpha, \boldsymbol{x}^{(0)} = \boldsymbol{0}, \boldsymbol{D}^{1/2}\boldsymbol{r}^{(0)} = \alpha\boldsymbol{e}_s$.
  2. When $t = 1$, we have

$$\boldsymbol{D}^{1/2}\boldsymbol{x}^{(1)} = \boldsymbol{D}^{1/2}\boldsymbol{x}^{(0)} - \boldsymbol{D}^{1/2}\boldsymbol{\nabla}f(\boldsymbol{x}^{(0)})$$
$$= \alpha\boldsymbol{e}_s$$
$$\boldsymbol{D}^{1/2}\boldsymbol{r}^{(1)} = \alpha\boldsymbol{e}_s - (\boldsymbol{I} - (1-\alpha)\boldsymbol{A}\boldsymbol{D}^{-1})\boldsymbol{D}^{1/2}\boldsymbol{x}^{(1)}$$
$$= \alpha(1-\alpha)\boldsymbol{A}\boldsymbol{D}^{-1}\boldsymbol{e}_s.$$

  3. When $t \geq 1$, we have

$$\delta_{t+1} = \left(\tfrac{2}{1-\alpha} - \delta_t\right)^{-1}$$
$$\hat{\boldsymbol{f}}^{(t)} = \tfrac{2\delta_{t+1}}{1-\alpha}\boldsymbol{D}^{1/2}\boldsymbol{r}^{(t)} + \delta_t\delta_{t+1}\boldsymbol{D}^{1/2}\boldsymbol{\Delta}^{(t)}$$
$$\boldsymbol{f}^{(t+1)} = \boldsymbol{f}^{(t)} + \hat{\boldsymbol{f}}^{(t)}$$
$$\boldsymbol{D}^{1/2}\boldsymbol{r}^{(t+1)} = \boldsymbol{D}^{1/2}\boldsymbol{r}^{(t)} - \left(\boldsymbol{I} - (1-\alpha)\boldsymbol{A}\boldsymbol{D}^{-1}\right)\hat{\boldsymbol{f}}^{(t)}.$$

  For LocalCH, we change the update $\hat{\boldsymbol{f}}^{(t)}$ to the update $\hat{\boldsymbol{f}}_{\mathcal{S}_t}^{(t)}$. The final estimate $\hat{\boldsymbol{f}} := \boldsymbol{D}^{1/2}\boldsymbol{x}^{(T)}$.

- **Chebyshev for Katz.** We want to solve $(\boldsymbol{I} - \alpha\boldsymbol{A})\boldsymbol{x} = \boldsymbol{e}_s$. We assume $\mu = 1 - \alpha\lambda_{\max}(\boldsymbol{A}), L = 1 - \alpha\lambda_{\min}(\boldsymbol{A})$. Then, the updates of LocalCH for Katz centrality are

  1. When $t = 0, \boldsymbol{x}^{(0)} = \boldsymbol{0}, \boldsymbol{r}^{(0)} = \boldsymbol{e}_s$. To obtain an initial value $\delta_1$, we have $\delta_1 = \alpha d_{\max}$ [7]

---

[7]This is because $|\lambda(\boldsymbol{A})| \leq d_{\max}$.

2. When $t = 1$, we have

$$\boldsymbol{x}^{(1)} = \boldsymbol{x}^{(0)} + \frac{2}{L+\mu}\boldsymbol{r}^{(0)} = \frac{2}{L+\mu}\boldsymbol{e}_s$$

$$\boldsymbol{r}^{(1)} = \boldsymbol{b} - (\boldsymbol{I} - \alpha\boldsymbol{A})\boldsymbol{x}^{(1)}$$

$$= \boldsymbol{e}_s - (\boldsymbol{I} - \alpha\boldsymbol{A})\frac{2}{L+\mu}\boldsymbol{e}_s, \boldsymbol{r}^{(1)}$$

$$= (1 - \tfrac{2}{L+\mu})\boldsymbol{e}_s + \frac{2\alpha}{L+\mu}\boldsymbol{A}\boldsymbol{e}_s.$$

3. When $t \geq 1$, it updates the estimate-residual pair as

$$\delta_{t+1} = \left(\tfrac{2}{\alpha d_{\max}} - \delta_t\right)^{-1}$$

$$\hat{\boldsymbol{x}}^{(t)} = \tfrac{4\delta_{t+1}}{L-\mu}\boldsymbol{r}^{(t)} + \delta_t\delta_{t+1}\boldsymbol{\Delta}^{(t)}$$

$$\boldsymbol{x}^{(t+1)} = \boldsymbol{x}^{(t)} + \hat{\boldsymbol{x}}^{(t)}$$

$$\boldsymbol{r}^{(t+1)} = \boldsymbol{r}^{(t)} - (\boldsymbol{I} - \alpha\boldsymbol{A})\hat{\boldsymbol{x}}^{(t)}.$$

For LocalCH, we change the update $\hat{\boldsymbol{x}}^{(t)}$ to the update $\hat{\boldsymbol{x}}_{\mathcal{S}_t}^{(t)}$. The final estimate is then $\hat{\boldsymbol{f}} := \hat{\boldsymbol{x}}^{(T)} - \boldsymbol{e}_s$.

We omit the details of LocalSOR and LocalGD for PPR and Katz as they can directly follow from (6) and (10), respectively. We implement all our proposed local solvers via the FIFO Queue data structure.

## B.6 FIFO-QUEUE and PRIORITY-QUEUE

We examine the different data structures of local methods and explore two types: the First-In-First-Out (FIFO) Queue, which requires constant time $\mathcal{O}(1)$ for updates, and the Priority Queue, which is used to implement the Gauss-Southwell algorithm as described in [48, 7]. We test the number of operations needed using these two data structures on five small graphs, including CORA ($n = 19793, m = 126842$), CORA-ML ($n = 2995, m = 16316$), CITESEER ($n = 4230, m = 10674$), DBLP ($n = 17716, m = 105734$), and PUBMED ($n = 19717, m = 88648$) as used in [9] and downloaded from https://github.com/abojchevski/graph2gauss.

Figure 10 presents the ratio of the total number of operations needed by APPR-FIFO and APPR-Priority-Queue. We tried four different settings; the performance of one does not dominate the other, and the Priority Queue is suitable for some nodes but not all nodes.

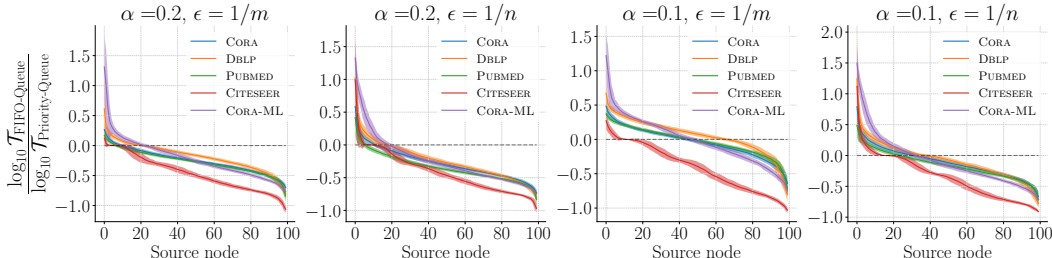

Figure 10: The number of operations ratio between the APPR (FIFO-Queue) and Gauss-Southwell (Priority-Queue). The number of operations of APPR is defined as $\mathcal{T}_{\text{FIFO-Queue}} = \sum_{t=0}^{T-1}\mathcal{S}_t$, $\mathcal{T}_{\text{Priority-Queue}} = \sum_{k=0}^{K}\left(d_{u_k} + \log_2|\operatorname{supp}(\boldsymbol{r}^{(k)})|\right)$ where $K$ is the total number of push operations used in Gauss-Southwell iteration and $\operatorname{supp}(\boldsymbol{r}^{(k)}) = \{u : r_u^{(k)} \neq 0, u \in \mathcal{V}\}$ and $\log_2|\operatorname{supp}(\boldsymbol{r}^{(k)})|$ is the number of operations needed by the priority queue for maintaining all residuals in $\boldsymbol{r}^{(k)}$.

## B.7 LocalGS(Dynamic) and LocalSOR(Dynamic)

The dynamic PPR algorithm based on LocalGS is presented in Algo. 2 and 3, initially proposed in [73]. Specifically, Algo. 2 and 3 are the components of LocalGS(Dynamic) for updating the edge event $(u, v)$ while Algo. 4 and 5 are the components of LocagSOR(Dynamic) for update the edge event $(u, v)$. In practice, we follow the batch updates strategy proposed in InstantGNN [75], which we call Algo. 3 and 5 for list of edge events then call Algo. 2 and 4 once after this batch update of edge events.

---

**Algo 2** LocalGS$(\mathcal{G}, \boldsymbol{p}_s, \boldsymbol{r}_s, \epsilon, \alpha, s)$

1: **while** $\max_u r_s[u] \geq \epsilon d_{\text{out}}[u]$ **do**
2:     Push$(u)$
3: **while** $\min_u r_s[u] \leq -\epsilon d_{\text{out}}[u]$ **do**
4:     Push$(u)$
5: **return** $(\boldsymbol{p}_s, \boldsymbol{r}_s)$

6: **procedure** Push$(u)$:
7:     $p_s[u] \leftarrow p_s[u] + r_s[u]$
8:     **for** $v$ in $\mathcal{N}_{\text{out}}(u)$ **do**
9:     $r_s[u] \leftarrow 0$
10:      $r_s[v] \leftarrow r_s[v] + \frac{(1-\alpha) \cdot r_s[u]}{d_{\text{out}}[u]}$

---

**Algo 3** LOCALGS(DYNAMIC)$(\mathcal{G}, \boldsymbol{p}_s, \boldsymbol{r}_s, \epsilon, \alpha, s, (u, v))$

1: Apply Insert/Delete of $(u, v)$ to $(\boldsymbol{p}_s, \boldsymbol{r}_s)$.
2: **return** LocalGS$(\mathcal{G}, \boldsymbol{p}_s, \boldsymbol{r}_s, \epsilon, \alpha, s)$

3: **procedure** Insert$(u, v)$
4:     $p_s[u] \leftarrow p_s[u] * d_{\text{out}}[u]/(d_{\text{out}}[u] - 1)$
5:     $r_s[u] \leftarrow r_s[u] - p_s[u]/d_{\text{out}}[u]$
6:     $r_s[v] \leftarrow r_s[v] + (1 - \alpha) \cdot p_s[u]/d_{\text{out}}[u]$

7: **procedure** Delete$(u, v)$
8:     $p_s[u] \leftarrow p_s[u] * d_{\text{out}}[u]/(d_{\text{out}}[u] + 1)$
9:     $r_s[u] \leftarrow r_s[u] + p_s[u]/d_{\text{out}}[u]$
10:     $r_s[v] \leftarrow r_s[v] - (1 - \alpha) \cdot p_s[u]/d_{\text{out}}[u]$

---

**Algo 4** LocalSOR$(\mathcal{G}, \boldsymbol{p}_s, \boldsymbol{r}_s, \epsilon, \alpha, s, \omega)$

1: **while** $\max_u r_s[u] \geq \epsilon d_{\text{out}}[u]$ **do**
2:     Push$(u)$
3: **while** $\min_u r_s[u] \leq -\epsilon d_{\text{out}}[u]$ **do**
4:     Push$(u)$
5: **return** $(\boldsymbol{p}_s, \boldsymbol{r}_s)$

6: **procedure** Push$(u)$:
7:     $p_s[u] \leftarrow p_s[u] + \omega r_s[u]$
8:     $r_s[u] \leftarrow r_s[u] - \omega r_s[u]$
9:     **for** $v$ in $\mathcal{N}_{\text{out}}(u)$ **do**
10:      $r_s[v] \leftarrow r_s[v] + \omega \cdot \frac{(1-\alpha) \cdot r_s[u]}{d_{\text{out}}[u]}$

---

**Algo 5** LOCALSOR(DYNAMIC)$(\mathcal{G}, \boldsymbol{p}_s, \boldsymbol{r}_s, \epsilon, \alpha, \omega, s, (u, v))$

1: Apply Insert/Delete of $(u, v)$ to $(\boldsymbol{p}_s, \boldsymbol{r}_s)$.
2: **return** LocalSOR$(\mathcal{G}, \boldsymbol{p}_s, \boldsymbol{r}_s, \epsilon, \alpha, s, \omega)$

3: **procedure** Insert$(u, v)$
4:     $p_s[u] \leftarrow p_s[u] * d_{\text{out}}[u]/(d_{\text{out}}[u] - 1)$
5:     $r_s[u] \leftarrow r_s[u] - p_s[u]/d_{\text{out}}[u]$
6:     $r_s[v] \leftarrow r_s[v] + (1 - \alpha) \cdot p_s[u]/d_{\text{out}}[u]$

7: **procedure** Delete$(u, v)$
8:     $p_s[u] \leftarrow p_s[u] * d_{\text{out}}[u]/(d_{\text{out}}[u] + 1)$
9:     $r_s[u] \leftarrow r_s[u] + p_s[u]/d_{\text{out}}[u]$
10:     $r_s[v] \leftarrow r_s[v] - (1 - \alpha) \cdot p_s[u]/d_{\text{out}}[u]$

---

## B.8 InstantGNN(LocalGS) and InstantGNN(LcalSOR)

We present InstantGNN(LocalGS) and InstantGNN(LcalSOR) in Algo. 6 and Algo. 7, respectively. Additional parameter $\beta$ is used; in practice, we choose $\beta = 1/2$ for all our experiments.

---

**Algo 6** InstantGNN(LocalGS)$(\mathcal{G}, \boldsymbol{p}, \boldsymbol{r}, \epsilon, \alpha, s, \beta)$

1: **while** $\max_u |r[u]| \geq \epsilon d_{\text{out}}^{1-\beta}[u]$ **do**
2:     Push$(u)$
3: **return** $(\boldsymbol{p}, \boldsymbol{r})$

4: **procedure** Push$(u)$:
5:     $p[u] \leftarrow p[u] + \alpha \cdot r[u]$
6:     $r[u] \leftarrow 0$
7:     **for** $v$ in $\mathcal{N}_{\text{out}}(u)$ **do**
8:      $r_s[v] \leftarrow r[v] + \frac{(1-\alpha) \cdot r[u]}{d_{\text{out}}^{1-\beta}[u] d_{\text{out}}^{\beta}[v]}$

---

**Algo 7** InstantGNN(LocalSOR)$(\mathcal{G}, \boldsymbol{p}, \boldsymbol{r}, \epsilon, \alpha, s, \beta, \omega)$

1: **while** $\max_u |r[u]| \geq \epsilon d_{\text{out}}^{1-\beta}[u]$ **do**
2:     Push$(u)$
3: **return** $(\boldsymbol{p}, \boldsymbol{r})$

4: **procedure** Push$(u)$:
5:     $p[u] \leftarrow p[u] + \alpha \cdot \omega \cdot r[u]$
6:     $r[u] \leftarrow r[u] - \omega \cdot r[u]$
7:     **for** $v$ in $\mathcal{N}_{\text{out}}(u)$ **do**
8:      $r[v] \leftarrow r[v] + \omega \cdot \frac{(1-\alpha) \cdot r[u]}{d_{\text{out}}^{1-\beta}[u] d_{\text{out}}^{\beta}[v]}$

---

Table 3: Dataset Statistics sorted by $n + m$

| Dataset ID | Dataset Name | n | m | n + m |
|---|---|---|---|---|
| $\mathcal{G}_1$ | Citeseer | 3279 | 9104 | 12383 |
| $\mathcal{G}_2$ | Cora | 2708 | 10556 | 13264 |
| $\mathcal{G}_3$ | pubmed | 19717 | 88648 | 108365 |
| $\mathcal{G}_4$ | ogbn-arxiv | 169343 | 2315598 | 2484941 |
| $\mathcal{G}_5$ | com-youtube | 1134890 | 5975248 | 7110138 |
| $\mathcal{G}_6$ | wiki-talk | 2388953 | 9313364 | 11702317 |
| $\mathcal{G}_7$ | as-skitter | 1694616 | 22188418 | 23883034 |
| $\mathcal{G}_8$ | cit-patent | 3764117 | 33023480 | 36787597 |
| $\mathcal{G}_9$ | ogbl-ppa | 576039 | 42463552 | 43039591 |
| $\mathcal{G}_{10}$ | ogbn-mag | 1939743 | 42182144 | 44121887 |
| $\mathcal{G}_{11}$ | ogbn-proteins | 132534 | 79122504 | 79255038 |
| $\mathcal{G}_{12}$ | soc-lj1 | 4843953 | 85691368 | 90535321 |
| $\mathcal{G}_{13}$ | reddit | 232965 | 114615892 | 114848857 |
| $\mathcal{G}_{14}$ | ogbn-products | 2385902 | 123612606 | 125998508 |
| $\mathcal{G}_{15}$ | com-orkut | 3072441 | 234370166 | 237442607 |
| $\mathcal{G}_{16}$ | wiki-en21 | 6216199 | 321647594 | 327863793 |
| $\mathcal{G}_{17}$ | ogbn-papers100M | 111059433 | 3228123868 | 3339183301 |
| $\mathcal{G}_{18}$ | com-friendster | 65608366 | 3612134270 | 3677742636 |

## C   Datasets and More Experimental Results

### C.1   Datasets Collected

We collected 18 graph datasets listed in Table 3. To test LocalSOR on GNN propagation, we use three graphs with feature data from [75].

### C.2   Detailed experimental setups.

For all experiments, the maximum runtime limit is set to 14,400 seconds. All methods will terminate when this time limit is reached. Specifically, we use the following parameter settings for drawing Figure 1 and Figure 3.

- **Experimental settings for Figure 1.** For computing PPR, we use a high precision parameter of $\epsilon = 10^{-10}/m$ and set $\alpha_{\text{PPR}} = 0.1$. For calculating Katz centrality, we use $\alpha_{\text{Katz}} = 1/(|\boldsymbol{A}|_2 + 1)$. The temperature for the Heat Kernel equation is set to $\tau = 10$.

- **Experimental settings Figure 3.** We use the *wiki-talk* graph and $\alpha = 0.1$ for PPR calculation.

Table 4: **Parameters of InstantGNN**

| Dataset | $\beta$ | $\alpha$ | $\epsilon$ | lr | batchsize | dropout | hidden size | layers |
|---|---|---|---|---|---|---|---|---|
| ogbn-arxiv | 0.5 | 0.1 | 1e-7 | 1e-4 | 8192 | 0.3 | 1024 | 4 |
| ogbn-products | 0.5 | 0.1 | 1e-8 | 1e-4 | 10000 | 0.5 | 1024 | 4 |

Table 4 presents parameter settings of InstantGNN model training in our experiments. The GDE in InstantGNN is defined as $\boldsymbol{f} = \sum_{k=0}^{\infty} \alpha(1-\alpha)^k (\boldsymbol{D}^{\beta} \boldsymbol{A} \boldsymbol{D}^{1-\beta})^k \boldsymbol{x}$. It is the same as APPNP [30] when we set $\beta = 0.5$.

### C.3   More experimental results

Figure 11 presents the results of GPU-implemented methods for Katz. Table 5-7 presents more details of participation ratios for PPR, Katz, and HK. The ratios in tables are without normalization. Figure 12-15 present more results of Figure 6 on different datasets and settings. Figure 16 and 17 present dynamic PPR approximating and training GNN model results on different datasets.

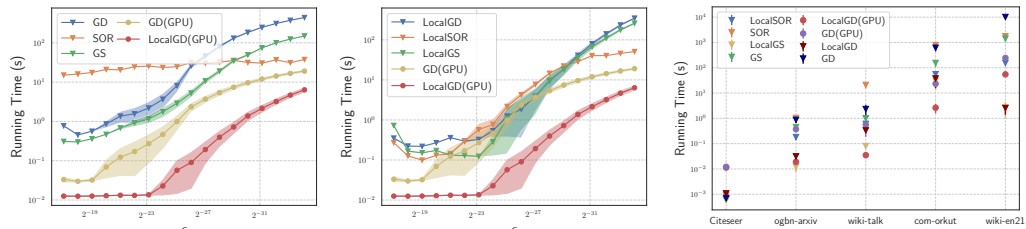

Figure 11: Comparison of running time (seconds) for CPU and GPU implementations for Katz.

Table 5: **Unnormalized participation ratios for PPR ($\alpha = 0.1$)**

| Graph | Vertices | Avg.Deg. | Participation Ratios | | | |
|---|---|---|---|---|---|---|
| | | | Min | Mean | Median | Max |
| Cora | 2708 | 3.90 | 1.11 | 2.57 | 2.60 | 5.68 |
| Citeseer | 3279 | 2.78 | 1.14 | 2.76 | 2.63 | 5.22 |
| pubmed | 19717 | 4.50 | 1.04 | 2.10 | 2.12 | 4.19 |
| ogbn-proteins | 132534 | 597.00 | 1.00 | 1.26 | 1.01 | 2.01 |
| ogbn-arxiv | 169343 | 13.67 | 1.03 | 1.84 | 1.65 | 4.12 |
| reddit | 232965 | 491.99 | 1.00 | 1.27 | 1.03 | 2.25 |
| ogbl-ppa | 576039 | 73.72 | 1.00 | 1.34 | 1.06 | 2.88 |
| com-youtube | 1134890 | 5.27 | 1.01 | 1.85 | 1.99 | 3.39 |
| as-skitter | 1694616 | 13.09 | 1.04 | 2.02 | 2.19 | 4.99 |
| ogbn-mag | 1939743 | 21.75 | 1.01 | 1.70 | 1.57 | 2.84 |
| ogbn-products | 2385902 | 51.81 | 1.00 | 1.46 | 1.19 | 2.84 |
| wiki-talk | 2388953 | 3.90 | 1.00 | 1.61 | 1.68 | 2.87 |
| com-orkut | 3072441 | 76.28 | 1.00 | 1.29 | 1.05 | 2.19 |
| cit-patent | 3764117 | 8.77 | 1.01 | 1.67 | 1.55 | 2.99 |
| soc-lj1 | 4843953 | 17.69 | 1.00 | 1.68 | 1.73 | 3.73 |
| wiki-en21 | 6216199 | 51.74 | 1.00 | 1.34 | 1.11 | 2.22 |
| com-friendster | 65608366 | 55.06 | 1.00 | 1.46 | 1.20 | 2.23 |
| ogbn-papers100M | 111059433 | 29.10 | 1.00 | 1.45 | 1.19 | 2.15 |

Table 6: **Unnormalized participation ratios for Katz($\alpha = 1/(1 + \|\boldsymbol{A}\|_2)$)**

| Graph | Vertices | Avg.Deg. | Participation Ratios | | | |
|---|---|---|---|---|---|---|
| | | | Min | Mean | Median | Max |
| Cora | 2708 | 3.90 | 1.01 | 7.62 | 3.34 | 30.24 |
| Citeseer | 3279 | 2.78 | 1.01 | 7.03 | 2.09 | 40.77 |
| pubmed | 19717 | 4.50 | 1.01 | 25.71 | 2.23 | 173.21 |
| ogbn-proteins | 132534 | 597.00 | 1.00 | 2181.89 | 2997.35 | 3397.70 |
| ogbn-arxiv | 169343 | 13.67 | 1.00 | 18.06 | 6.04 | 47.12 |
| reddit | 232965 | 491.99 | 1.00 | 3364.93 | 4412.49 | 4757.13 |
| ogbl-ppa | 576039 | 73.72 | 1.00 | 507.92 | 43.88 | 2984.20 |
| com-youtube | 1134890 | 5.27 | 1.00 | 10.86 | 8.65 | 42.78 |
| as-skitter | 1694616 | 13.09 | 1.00 | 52.56 | 5.07 | 702.30 |
| ogbn-mag | 1939743 | 21.75 | 1.00 | 8.24 | 6.71 | 182.96 |
| ogbn-products | 2385902 | 51.81 | 1.00 | 100.53 | 27.33 | 649.78 |
| wiki-talk | 2388953 | 3.90 | 1.00 | 404.15 | 587.18 | 701.41 |
| com-orkut | 3072441 | 76.28 | 1.00 | 637.67 | 50.15 | 1824.24 |
| cit-patent | 3764117 | 8.77 | 1.00 | 37.49 | 6.01 | 315.39 |
| soc-lj1 | 4843953 | 17.69 | 1.00 | 358.72 | 5.00 | 2122.33 |
| wiki-en21 | 6216199 | 51.74 | 1.00 | 110.88 | 156.60 | 189.79 |
| com-friendster | 65608366 | 55.06 | 1.00 | 8475.52 | 9.02 | 36776.54 |
| ogbn-papers100M | 111059433 | 29.10 | 1.00 | 97.53 | 7.93 | 726.71 |

Table 7: **Unnormalized participation ratios for HK($\tau = 10$)**

| Graph | Vertices | Avg.Deg. | Participation Ratios | | | |
| | | | Min | Mean | Median | Max |
|---|---|---|---|---|---|---|
| cora | 2708 | 3.90 | 1.65 | 8.43 | 6.62 | 34.85 |
| citeseer | 3279 | 2.78 | 1.82 | 7.69 | 5.81 | 35.71 |
| pubmed | 19717 | 4.50 | 1.15 | 26.02 | 7.09 | 242.95 |
| ogbn-proteins | 132534 | 597.00 | 437.31 | 5743.38 | 5712.33 | 12874.41 |
| ogbn-arxiv | 169343 | 13.67 | 1.95 | 15.48 | 10.61 | 109.58 |
| reddit | 232965 | 491.99 | 30.26 | 2697.53 | 2468.45 | 5395.28 |
| ogbl-ppa | 576039 | 73.72 | 2.02 | 1052.49 | 766.77 | 3860.21 |
| com-youtube | 1134890 | 5.27 | 1.06 | 8.66 | 4.73 | 58.02 |
| as-skitter | 1694616 | 13.09 | 1.51 | 13.04 | 7.18 | 45.05 |
| ogbn-mag | 1939743 | 21.75 | 1.50 | 2.63 | 1.96 | 16.90 |
| ogbn-products | 2385902 | 51.81 | 2.16 | 124.51 | 72.87 | 775.54 |
| wiki-talk | 2388953 | 3.90 | 1.00 | 8.19 | 1.43 | 65.72 |
| com-orkut | 3072441 | 76.28 | 6.02 | 655.85 | 367.02 | 7942.79 |
| cit-patent | 3764117 | 8.77 | 1.76 | 30.51 | 21.03 | 273.80 |
| soc-lj1 | 4843953 | 17.69 | 2.29 | 59.08 | 24.23 | 573.48 |
| wiki-en21 | 6216199 | 51.74 | 3.00 | 34.05 | 32.49 | 164.69 |
| com-friendster | 65608366 | 55.06 | 2.05 | 323.12 | 83.67 | 3092.69 |
| ogbn-papers100M | 111059433 | 29.10 | 1.44 | 51.49 | 21.40 | 275.48 |

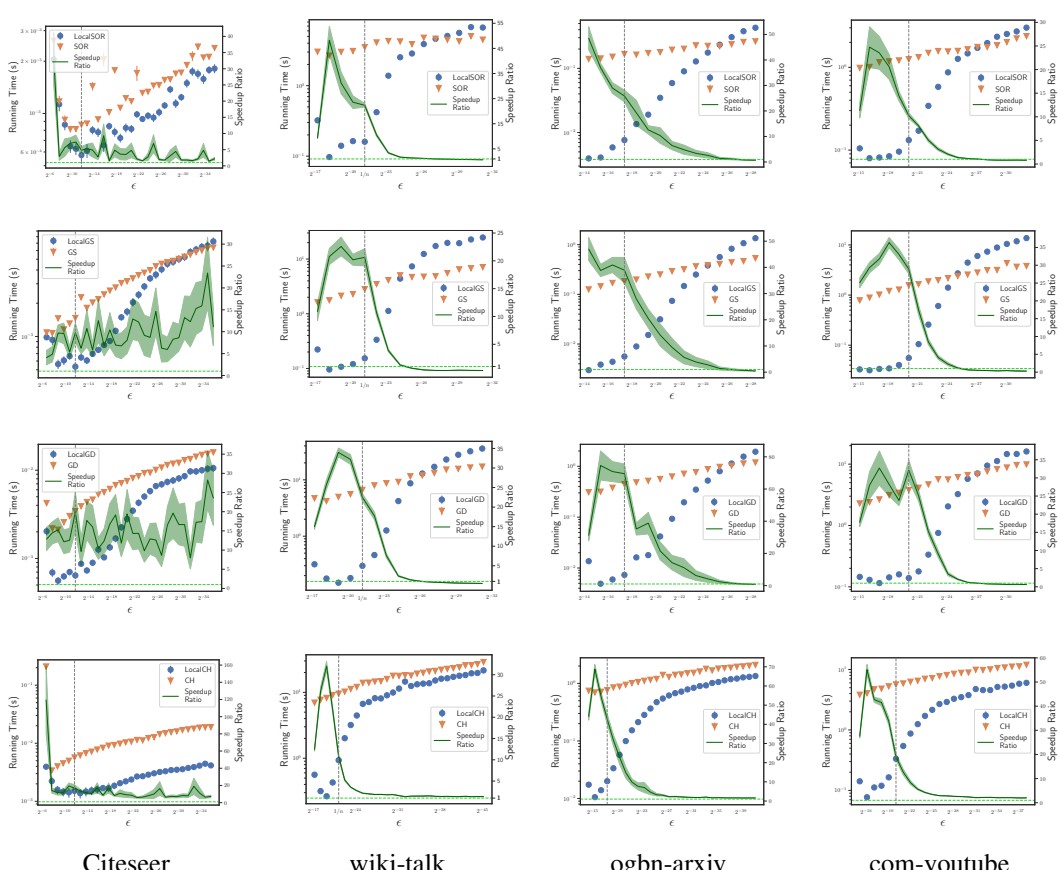

Figure 12: Running time (seconds) as a function of $\epsilon$ for PPR on several datasets with $\alpha = 0.1$.

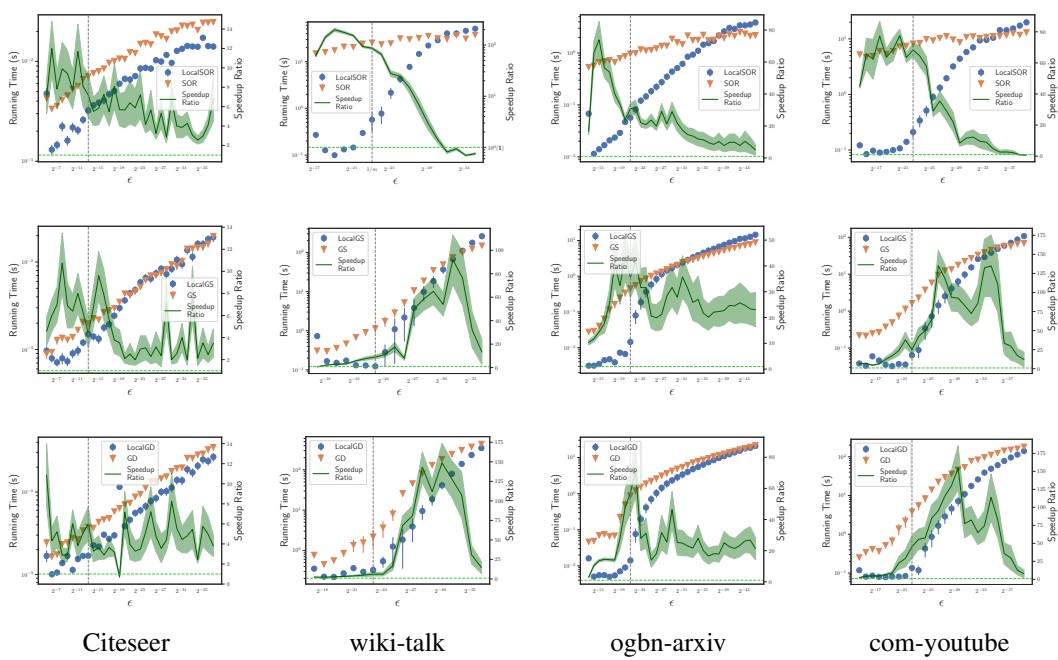

Figure 13: Running time (seconds) as a function of $\epsilon$ for Katz on several datasets with $\alpha = 1/(\|\boldsymbol{A}\|_2 + 1)$.

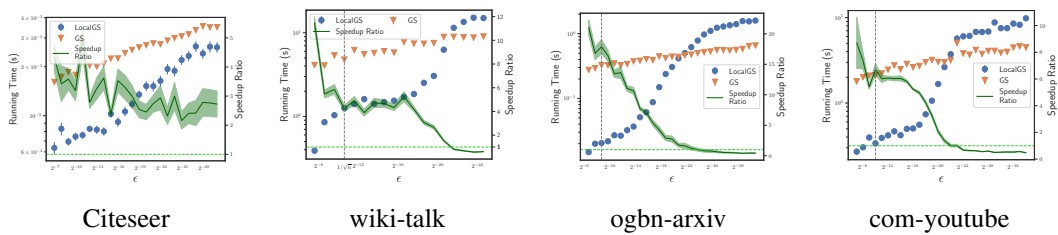

Figure 14: Running time (seconds) as a function of $\epsilon$ for HK on several datasets with $\tau = 10$.

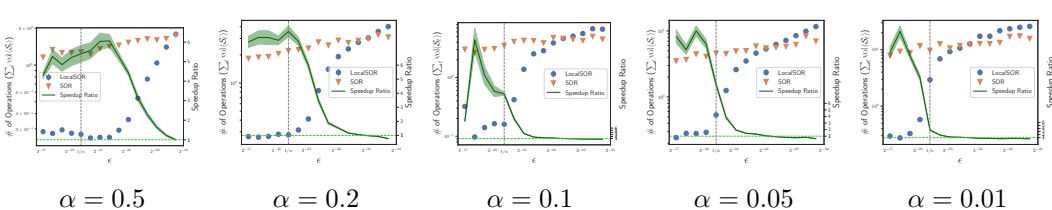

Figure 15: Running time (seconds) of SOR and LocalSOR as a function of $\epsilon$ for PPR on the *wiki-talk* dataset with different $\alpha$.

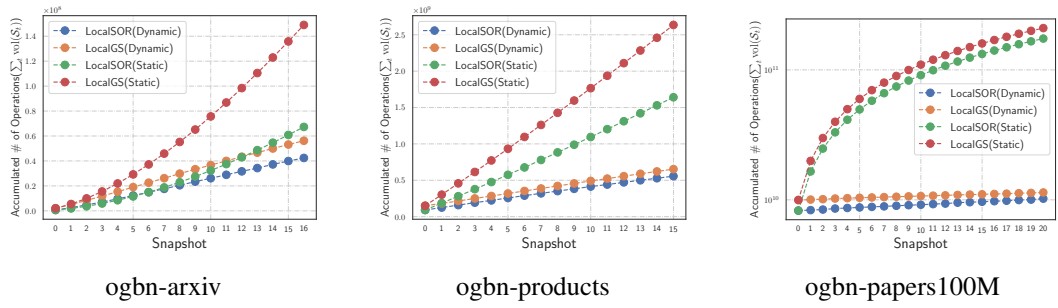

ogbn-arxiv             ogbn-products             ogbn-papers100M

Figure 16: Accumulated number of operations on some dynamic graphs.

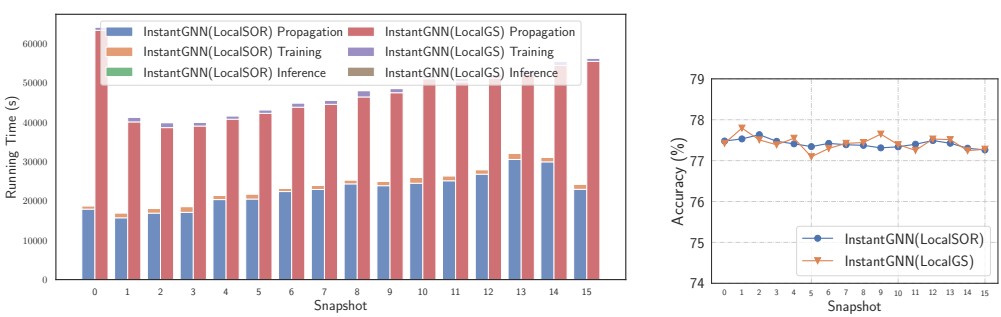

Figure 17: The performance of InstantGNN using SOR and GS propagation on ogbn-products.

## C.4 The efficiency of local methods on different types of graphs (tested on grid graphs).

Our local algorithms do not depend on graph types. However, our key assumption is that diffusion vectors are highly localizable, measured by the participation ratio. As demonstrated in Figure 1, almost all diffusion vectors have low participation ratios collected from 18 real-world graphs. These graphs are diverse, ranging from citation networks and social networks to gene structure graphs. To further illustrate this, we conducted experiments where the graphs are grid graphs, and the diffusion vectors have high participation ratios (about $3.56 \times 10^{-6}$, with a grid size of 1000x1000, i.e., $10^6$ nodes and 1,998,000 edges). We set $\alpha_{PPR} = 0.1$. The figure below presents the running time as a function of $\epsilon$ over a grid graph with 50 sampled source nodes.

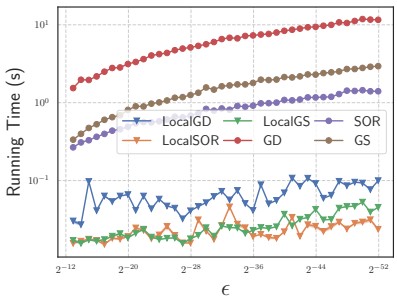

The results indicate that local solvers are more efficient than global ones even when $\epsilon$ is very small.

