# OpenReview forum: "Faster Local Solvers for Graph Diffusion Equations"
_NeurIPS.cc/2024/Conference — NeurIPS 2024 poster_

### Official Review · Reviewer_X6sG · 2024-07-08

**Soundness:** 3
**Presentation:** 3
**Contribution:** 3
**Rating:** 5
**Confidence:** 2

**Summary:**

The paper, "Faster Local Solvers for Graph Diffusion Equations," addresses the efficiency of computing Graph Diffusion Equations (GDEs) such as Personalized PageRank (PPR), Katz centrality, and the Heat kernel, which are essential for various graph-related problems like clustering and training neural networks. Traditional methods for solving GDEs are computationally intensive for large-scale graphs. This paper introduces a novel framework for approximating GDEs using local diffusion processes, which significantly reduces computational time and improves scalability by leveraging the localization property of diffusion vectors. The proposed local solvers are highly parallelizable and suitable for GPU implementation, offering up to a hundred-fold speed improvement and applicability to large-scale dynamic graphs. The paper also discusses the potential for these methods to enhance local message-passing mechanisms in Graph Neural Networks (GNNs).

**Strengths:**

The introduction of a novel framework for localizing the computation of GDEs using local diffusion processes is a significant contribution. This approach reveals the suboptimality of existing local solvers and provides a more efficient solution.


The paper offers a solid theoretical foundation, proving that popular diffusion vectors have strong localization properties using the participation ratio. It demonstrates that Approximate Personalized PageRank (APPR) can be treated as a special case of the proposed framework, providing better diffusion-based bounds. The design of simple and fast local methods based on standard gradient descent and the local Chebyshev method for symmetric propagation matrices is well-founded.


Experimental results on GDE approximation for PPR, HK, and Katz show significant acceleration over standard methods. The proposed local solvers demonstrate up to a hundred-fold speed improvement. The paper also shows that the new methods can be naturally adopted to approximate dynamic diffusion vectors, and they outperform standard PPR-based GNNs in training speed.

**Weaknesses:**

Precision Limitations: While the paper shows significant speedups for lower precision settings, the performance gain diminishes as higher precision is required. This limitation could affect the applicability of the methods in scenarios where high precision is crucial.

Sequential Nature: (Please correct if I am wrong.) Despite improvements, the reliance on sequential updates in some methods (like LocalSOR) still poses challenges for achieving maximal parallel efficiency.

Complexity of Analysis: The runtime analysis for some of the proposed methods, such as the Local Chebyshev method, is noted as complex and remains an open problem. Maybe simplifying this analysis or providing more intuitive explanations could enhance the paper’s accessibility.

**Questions:**

What are the potential trade-offs between precision and computational efficiency, and how can practitioners balance these in practical applications?

The paper includes quite some datasets to test, with biggest of ogbn-papers100M. How does the proposed method handle very large-scale dynamic graphs in real-time applications, and what are the potential bottlenecks?

**Limitations:**

See above

---

> ### Author Rebuttal · Authors · 2024-08-04
>
> Thank you for taking the time and effort to review our paper. We appreciate your positive perspective. We resolve your concerns as follows.
>
> ---
>
> **Q1.** What are the potential trade-offs between precision and computational efficiency, and how can practitioners balance these in practical applications? The performance gain diminishes as higher precision is required, which could affect the methods' applicability in scenarios where high precision is crucial.
>
> **A:** Please see our general response to this concern. Furthermore, it is true that when $\epsilon$ is lower, the run time of our local methods is getting closer to global ones. In the large range of precision $\epsilon \leq 10^{-4}/n$, however, they are still competitive and use less time. Furthermore, high precision is not required in most empirical settings of graph learning problems, including local clustering, training GNN models, and learning graph embeddings. It is still an open problem that the minimal complexity is significant when $\epsilon$ is lower ( proportional to $\log 1/\epsilon$). In terms of training dynamic GNNs, the precision is about $\mathcal{O}(1/m)$. In terms of local graph clustering, the precision is about $1/n$. The essential question is whether or not our methods are optimal or close to the optimal local methods.  We believe this question is worth studying further in our future work.
>
> ---
>
> **Q2.** Despite improvements, relying on sequential updates in some methods (like LocalSOR) still poses challenges for achieving maximal parallel efficiency.
>
> **A:** Yes, the sequential updates of LocalSOR are not suitable for parallelization. This is why we propose local gradient descent and the local Chebyshev method to speed up these methods further. However, we still think the local SOR is valuable when the GPU resources are not available and is more power efficient. In practice, when high precision is needed, we recommend using LocalSOR when GPUs are not available, as both LocalSOR and LocalCH have accelerated rates in the global setting.
>
> ---
>
> **Q3.** Comments on the runtime analysis for some proposed methods, such as the Local Chebyshev method.
>
> **A:** Thank you for this great point. We are still investigating this strategy. Please also see a more detailed explanation in our general response. The sublinear runtime analysis for LocalCH is complicated since it does not follow the monotonicity property during the updates. One cannot directly follow existing techniques. One possible solution is that the diffusion process can be effectively characterized by the residuals of LocalCH, which is a type of second-order difference equation with parameterized coefficients. We then use this second-order difference equation to provide a bound. However, the core difficulty is that this difference equation is difficult to analyze when the coefficients are parameterized. We are actively investigating this direction.
>
> ---
>
> **Q4.** How does the proposed method handle very large-scale dynamic graphs in real-time applications, and what are the potential bottlenecks?
>
> **A:** The potential bottleneck is memory usage when the precision is high. We tried to apply our local solvers to the ogbn-papers100M graph, but our GPU memory is limited to 24GB. It fails to train due to memory issues. One reason is that the graph size is too large. However, the other reason could be that memory usage is linear to $1/\epsilon$. We will investigate this further and explore more efficient ways to reduce memory usage in this case. However, our paper shows that speedup is very significant when the participation ratio is low for middle-scale dynamic graphs where we tested these local solvers on training dynamic GNN models on the ogbn-arxiv and ogbn-products datasets.
>
> **We are happy to discuss any further concerns you may have.**

---

### Official Review · Reviewer_X7w6 · 2024-07-13

**Soundness:** 3
**Presentation:** 3
**Contribution:** 2
**Rating:** 4
**Confidence:** 3

**Summary:**

This paper proposes a novel framework for approximately solving graph diffusion equations using a local diffusion process. In addition,  the proposed method can effectively localizes standard iterative solvers by designing simple and provably sublinear time algorithms.

**Strengths:**

+ The problem is well motivated and the paper is well-written.
+ Extensive experiments are conducted to showcase the efficiency of graph diffusion framework to approximating graph diffusion equations.
+ The paper provides a good summary of existing graph diffusion equations.
+ The authors provide code and details of implementations.

**Weaknesses:**

- In this paper, the authors explore 18 different graphs, can the authors show/provide which local GDE solver achieves better performance on which type(s) of graphs?
- Computational complexity/cost is missing.

**Questions:**

See comments in Weaknesses.

**Limitations:**

Not applicable.

---

> ### Author Rebuttal · Authors · 2024-08-02
>
> Thank you for taking the time and effort to review our paper. We believe there may have been some misunderstandings, and your two concerns can be effectively resolved as follows:
>
> ---
>
> **Q1.** Can the authors show/provide which local GDE solver achieves better performance on which type(s) of graphs?
>
> **A:** **We answered this question in Section 5.1 (Line 272 to Line 282).** Our results in the lower precision setting suggest that LocalSOR and LocalGD perform best overall. Figure 6 illustrates the performance of LocalSOR and LocalGS for PPR and Katz. As $\epsilon$ becomes smaller, the speedup of LocalSOR over LocalGS becomes more significant. In practice, when GPU resources are available, using LocalCH (corresponding to an accelerated convergence rate) is recommended instead of LocalGD in the high precision range. When GPU resources are unavailable, using LocalSOR (corresponding to an accelerated convergence rate with optimal $\omega$) instead of APPR is recommended in the high precision range.
>
> **More importantly, our local algorithms do NOT depend on graph types.** A fundamental assumption of our paper is that the computed diffusion vectors are highly localizable, measured by the participation ratio. As demonstrated in Figure 1, almost all diffusion vectors have low participation ratios collected from 18 real-world graphs. It is worth studying whether high-participation ratio diffusion vectors exist in real-world graphs. Interestingly, when the participation ratio is high, local solvers still offer some level of speedup compared to global ones. To verify this, we conducted a simple experiment where the graphs are grid graphs, and the diffusion vectors have high participation ratios. The results indicate that local solvers are more efficient than global ones, even when the participation ratios are high. **Please check our attached PDF file to see the detailed experimental results on grid graphs.**
>
> ---
>
> **Q2.** Computational complexity/cost is missing.
>
> **A:** **There seems to be some misunderstanding. The analysis of the computational complexity of local solvers is actually one of our core contributions.** The main contribution of our paper is to show the local runtime complexity of these two methods for different GDEs. Our framework is quite general and applicable to a broad range of GDEs. Specifically, these runtime complexities are stated in Theorem 3.3 (LocalSOR for PPR), Corollary 3.4 (LocalSOR for Katz), and Corollary 3.6 (LocalGD for PPR and Katz). Therefore, we are unsure whether you mean the runtime complexity of LocalSOR and LocalGD for Katz and PPR, or something else.
>
> **We are ready and happy to discuss any further concerns you may have.**

---

> > ### Comment · Reviewer_X7w6 · 2024-08-12
> > **Thank you.**
> >
> > I appreciate the author for the detailed response. After carefully reading the rebuttal, I am retaining my score.

---

> > > ### Author Response · Authors · 2024-08-12
> > > **Request for Clarification on Remaining Concerns (We are confused)**
> > >
> > > We sincerely appreciate your response and your careful reading of our rebuttal. However, we are still uncertain whether all your concerns have been sufficiently addressed. Based on your score, it seems there are reasons to reject the paper, such as limited evaluation, that outweigh the reasons to accept it, such as a good evaluation. But, currently, you only have one question and one misunderstanding point. So, we are confused.
> > >
> > > To facilitate a valid discussion, please specify any other concerns you might have so we can address them appropriately. Thank you.

---

### Official Review · Reviewer_JA1M · 2024-07-13

**Soundness:** 3
**Presentation:** 3
**Contribution:** 3
**Rating:** 7
**Confidence:** 2

**Summary:**

This paper proposes a new local iterative framework for solving graph diffusion equations (GDEs). Specifically, the framework approximates GDEs through a local diffusion process, leveraging the strong localization properties of diffusion vectors such as personalized PageRank, Katz centrality, and Heat Kernel. The proposed local solvers can achieve sublinear runtime complexity under certain monotonicity assumptions. Empirical results demonstrate that these solvers significantly accelerate their standard counterparts on several large-scale benchmark datasets.

**Strengths:**

1. The paper is well-structured and clear.

2. The theoretical analysis is rigorous and sound, providing runtime complexity bounds for some of the proposed local solvers, which are better than their standard counterparts.

3. The effectiveness of the proposed methods is well-supported by experimental results on large-scale benchmark datasets.

**Weaknesses:**

1. It is not clear how graph structures, such as sparsity and spectral properties, impact the runtime complexity of the local solvers. A discussion on the potential influence of graph structures on runtime complexity would be helpful.

2. The runtime complexity analysis assumes that the updates of local solvers satisfy monotonicity properties. However, LocalCH does not satisfy these properties during updates. Establishing runtime bounds for LocalCH may require different techniques.

3. In Figure 7, when $\epsilon \geq 2^{-31}$, the running time of LocalGD is the same as GD, indicating that LocalGD may not speed up the standard counterparts when $\epsilon$ is sufficiently small.

**Questions:**

Please refer to Weaknesses.

---

> ### Author Rebuttal · Authors · 2024-08-04
>
> We are happy that you like our work. We appreciate your time and effort in reviewing our submission. We addressed your concerns as follows:
>
> ---
>
> **Q1.** It is unclear how graph structures, such as sparsity and spectral properties, impact the runtime complexity of the local solvers.
>
> **A:** In terms of graph structure, our local algorithms do not depend on graph types. A key assumption of our paper is that the computed diffusion vectors are highly localizable, measured by the participation ratio. It is worth studying whether high-participation ratio diffusion vectors exist in real-world graphs. To verify this further, we conducted a simple experiment where the graphs are grid graphs, and the diffusion vectors have low participation ratios. The results indicate that local solvers are more efficient than global ones, even with low participation ratios. **Please also check our general response and our attached PDF file to see the detailed experimental results.**
>
> In terms of spectral properties, the performance of both local and global solvers is affected by the spectra of the underlying graph. For example, let $\lambda_1,\lambda_2, \ldots, \lambda_n$ be the eigenvalues of normalized Laplacian of the underlying matrix defined by the graph. Then, the condition number is determined by $\lambda_1$ and $\lambda_n$, which will dominate the convergence rate of all methods. Furthermore, when the eigenvalues are clustered together, all methods will be easier to converge due to the same reason. Thank you for this great point, and we will add discussions on this to our manuscript.
>
> ---
>
> **Q2.** Comments on the difficulty of accelerated bound analysis for LocalCH.
>
> **A:** The sublinear runtime analysis for LocalCH is complicated since it does not follow the monotonicity property during the updates. One cannot directly follow existing techniques. One possible solution is that the diffusion process can be effectively characterized by the residuals of LocalCH, which is a type of second-order difference equation with parameterized coefficients. We then use this second-order difference equation to provide a bound. However, the core difficulty is that this difference equation is difficult to analyze when the coefficients are parameterized.  We are actively investigating this direction.
>
> ---
>
> **Q3.** When $\epsilon$ is sufficiently small, LocalGD may not speed up the standard counterparts.
>
> **A:** We admit that when $\epsilon \rightarrow 0$, it is hopeless to speedup the standard counterparts as the optimal first-order methods need run time $\mathcal{O}(m/\sqrt{\alpha})$ for computing PPR vectors. In the high precision ( in Figure 7), when $\epsilon \geq 2^{-31}$, the running time of LocalGD is slightly less than that of GD. More importantly, the computation of high-precision diffusion vectors is not our main goal. We aim to develop faster local methods when high precision is not a strong requirement. Many applications of graph learning problems only need lower precisions, such as local clustering and training GNN models.
>
> To summarize, for the first time, we propose a novel framework for designing efficient local solvers that could be applied to many graph diffusion equations (GDEs). These GDEs solvers can be applied to local clustering and training graph neural networks, improving the efficiency of these algorithms. We are happy to have further discussions if needed!

---

> > ### Comment · Reviewer_JA1M · 2024-08-12
> >
> > Thank you for your response. I continue to recommend acceptance and retain my score.

---

### Official Review · Reviewer_kfsj · 2024-07-15

**Soundness:** 3
**Presentation:** 3
**Contribution:** 3
**Rating:** 6
**Confidence:** 4

**Summary:**

This paper proposes a suite of fast methods to approximately compute graph diffusion vectors such as Personalized PageRank, Katz centrality and the heat kernel. A notable feature of the proposed methods is that they are easily parallelizable and hence can achieve further acceleration on GPU. The authors also provide a running time bound for each method that they introduce. Empirical results show that the new local methods can achieve up to a hundred-fold speedup when compared to their global counterpart.

**Strengths:**

- The paper is well-written and easy to follow.
- The local diffusion framework is applicable to computing several important graph diffusion vectors.
- The experiments are reasonably comprehensive.

**Weaknesses:**

- When compared with APPR, the speedup is not really captured by Theorem 3.3 and Corollary 3.6. The authors only showed that $\overline{\mbox{vol}}(\mathcal{S}_T)/\bar\gamma_T \le 1/\epsilon$, but it requires strict inequality to achieve nontrivial speedup in terms of worse-case running time. It is not clear how tight this bound is in general. If the bound is tight, then in the worst case there is no speedup. The authors should comment on if there are classes of graphs over which there will be a notable gap between the 2 quantities $\overline{\mbox{vol}}(\mathcal{S}_T)/\bar\gamma_T$ and $1/\epsilon$, so that the result provides a meaningful improvement.
- Because the theorems concerning the worst-case running time do not seem to capture a clear improvement over simple baseline local methods, it is unclear where exactly the speedup reported in Table 2 comes from.
- In LocalGD and LocalCH, the authors did not provide an update rule (or even a definition) for $\mathcal{S}_t$. Since $\mathcal{S}_t$ is part of the local diffusion process, the authors should specify how $\mathcal{S}_t$ is updated or defined at each step. I guess that $\mathcal{S}_t$ depends on the termination condition, but I could not find where the authors mention about it in the main paper.

**Questions:**

- Line 188: The open problem is recently solved by Martínez-Rubio et al, Accelerated and Sparse Algorithms for Approximate Personalized PageRank and Beyond, 2023.
- When eps goes to 0, do the approximate solutions converge to the exact global solutions?

**Limitations:**

Yes

---

> ### Author Rebuttal · Authors · 2024-08-04
>
> Thank you for taking the time and effort to review our paper carefully. We appreciate your positive perspective on our paper. Your concerns and our responses are listed as follows.
>
> ---
>
> **Q1.** The concern of two quantities $ \overline{\operatorname{vol}}\left(\mathcal{S}_T\right) / \overline{\gamma}_T $ and $ 1 / \epsilon$: It is unclear how tight this bound is in general.
>
> **A:** Thank you for this great point. Our bound is slightly rough. The actual inequality is $ \overline{\operatorname{vol}}\left(\mathcal{S}_T\right) / \overline{\gamma} _T < 1 / \epsilon$ as long as $T \geq 1$.  The inequality used in Line 664 $(\leq)$ is actually $(<)$, i.e., $\sum _{i=1}^{ |\mathcal S_t|} r _{u_i} / ||\boldsymbol{r}^{(0)} ||_1 < \sum _{i=1}^{ |\mathcal S_t|} r _{u_i} / ||\boldsymbol{r}^{(t)} ||_1$ as long as $t \geq 1$. This strict inequality is achievable if the algorithm runs more than one iteration. The reason is that the monotonicity property of $|| \boldsymbol r^{(t)} ||_1$ is actually $|| \boldsymbol r^{(0)} ||_1 < || \boldsymbol r^{(1)} ||_1 <\cdots < ||  \boldsymbol r^{(t)} ||_1 < \cdots$ where there are at least some magnitudes that are moved from $\boldsymbol r^{(t)}$ to $\boldsymbol x^{(t)}$ per-iteration. Therefore, in the worst case, the improvement of our bound is significant if the factor $1/\epsilon$ is the main concern.
>
> More importantly, it is better to compare two local bounds presented in Theorem 3.3 instead of only considering two quantities. We conducted a simple experiment to compare these two quantities (please see our detailed experimental results in the attached file) and two local bounds. Our results indicate the effectiveness of our local bounds. The time-evolving bound is more suitable and mirrors the actual performance.
>
> ---
>
> **Q2.** It is unclear where exactly the speedup reported in Table 2 comes from.
>
> **A:** Table 2 presents the speedup ratio of computing PPR vectors comparing the local and global solvers where we set $\epsilon = 1/n$ and report the value $\mathcal{T} _{\mathcal A} / \mathcal{T} _{ \text{Local} \mathcal A} $ where $\mathcal{T} _{\mathcal A}$ is the total number of operations needed for algorithm $\mathcal{A}$. The results of different local solvers are in Figure 4. These results indicate that local solvers speed up significantly compared to their global counterparts.
>
> ---
>
> **Q3.** The definition of $\mathcal{S}_t$ in LocalGD and LocalCH is missing.
>
> **A:** Thank you for pointing out this. The active set $\mathcal{S} _t$ indeed depends on the termination condition. We defined $\mathcal{S}_t = \\{ u _1, u _2, \ldots, u _{\left| \mathcal{S} _t \right|} \\}$ be the set of active nodes processed in $t$ iteration in Section 3.1. Specifically, at any time $t$, $\mathcal S _t$ maintains the set $\\{ u \in \mathcal{V} : |r_u| \geq   \alpha \epsilon d_u \\}$ for calculating PPR. The definition of Katz's score is similar. We will clarify these $\mathcal{S} _t$ for all local solvers in our manuscript.
>
> ---
>
> **Q4.** For the PPR computation, the open problem was recently solved by Martínez-Rubio et al. (2023, COLT)
>
> **A:** Thank you for pointing out this latest work. We are also aware of Martínez-Rubio's work when we are submitting our manuscript. First of all, Algorithm 4 in Martínez-Rubio's work is proposed for solving PPR, not graph diffusion vectors (GDEs). But, the problem in our paper is solving GDEs which is more general than the open problem.
>
> When we empirically compared our work, LocalSOR, with ASPR, we found that ASPR is much slower than our proposed LocalSOR. **The main reason is that ASPR is based on a nested subspace pursuit strategy, and the corresponding iteration complexity is bounded by $\mathcal O (|\mathcal S^\star| / \sqrt{\alpha} )$ where $\mathcal S^\star$ is the support of the optimal solution. This bound deteriorates to $\tilde {\mathcal {O}} (n / \sqrt{ \alpha})$ when the solution is dense, with $ n$ representing the number of nodes in $\mathcal G$, which could be less favorable than that of standard solvers under similar conditions. Please see the detailed experimental results in our attached file.** Moreover, the nested computational structure provides a constant factor overhead, which could be significant in practice.
>
> ---
>
> **Q5.** When $\epsilon \rightarrow 0$, do $\boldsymbol{x}^{(t)}$ converge to $\boldsymbol{x}^{*}$?
>
> **A:** Yes. For PPR using LocalSOR in Theorem 3.3, we proved that the estimate $\boldsymbol{x}^{(T)}$ satisfies $
> ||\boldsymbol{D} ^{-1}(\boldsymbol{x} ^{(T)}-\boldsymbol{f} _{\text{PPR}})|| _{\infty} \leq \epsilon$.  This means $\boldsymbol{x} ^{(T)} \rightarrow \boldsymbol{f} _{\text{PPR}}$ when $\epsilon \rightarrow 0$.  For Katz using LocalSOR in Corollary 3.4, the estimate $\hat{\boldsymbol{f}} _{\text {Katz }} =\boldsymbol{x}^{(T)}-\boldsymbol{e} _s$ satisfies $||\hat{\boldsymbol{f}} _{\text {Katz }}-\boldsymbol{f} _{\text {Katz }} ||_2 \leq ||(\boldsymbol{I}-\alpha \boldsymbol{A})^{-1} \boldsymbol{D} ||_1 \cdot \epsilon$. This means $\hat{\boldsymbol{f}} _{\text {Katz }} \rightarrow \boldsymbol{f} _{\text{Katz}}$ when $\epsilon \rightarrow 0$. One can obtain similar results for LocalGD.
>
> ---
>
> We hope that our clarifications can address your concerns. **We are happy to have further discussions if needed!**

---

> > ### Comment · Reviewer_kfsj · 2024-08-13
> >
> > I'd like to thank the authors for their detailed responses. My questions have been properly addressed. I will increase my score.

---

### Author Rebuttal · Authors · 2024-08-04

**Our General Responses**

We thank all reviewers for their time and effort in carefully reading our paper. To address some general concerns, we included experimental results in the attached PDF file. Furthermore, some general concerns are worth to response as follows:

---

**Q1. Potential trade-offs between precision and computational efficiency: When high precision is required, i.e., $\epsilon \rightarrow 0$, the efficiency of local solvers diminishes (Reviewer kfsj, JA1M, X6sG).**

**A:** To clarify, the effective speedup of local solvers over the standard ones is when $\epsilon \leq 10^{-4}/n$ empirically. This already covers many downstream applications, including local clustering (where $\epsilon = 10^{-6}$ and $n\approx 10^{-6}$ see Fountoulakis's work in [1]) and training graph neural networks (where $\epsilon = 10^{-4}$ but $n \geq 10^{5}$ see Bojchevski's work on the PPRGo model in [2]), as far as we know. It is a reasonable observation that the efficiency of local solvers diminishes when $\epsilon \rightarrow 0$, as the number of iterations needed for both local and global solvers is proportional to $\log(1/\epsilon)$. These local solvers are more like their global counterparts. Specifically, for the optimal first-order global solvers, they require $\mathcal{O}(m \sqrt{\kappa} \log(1/\epsilon))$, where $m$ is the number of edges in the graph and $\kappa$ is the condition number (e.g., $\kappa = 1/\alpha$ when the graph diffusion equation is PPR). Similarly, for the local solvers, as proved in our Theorem 3.3, the number of *local iterations* is also proportional to $\log(1/\epsilon)$. It is unknown, but it remains interesting to see whether the runtime of *optimal first-order local methods* is proportional to $\log(1/\epsilon)$. We will study this problem in future work.

---

**Q2. Comments on the difficulty of accelerated bound analysis for LocalCH (Reviewer JA1M, X6sG)**

**A:** We are actively investigating this direction and have some preliminary results. The sublinear runtime analysis for LocalCH is complicated since it does not follow the monotonicity property during the updates. One cannot directly follow existing techniques. One possible solution (we are investigating it) is that the diffusion process can be effectively characterized by the residuals of LocalCH, which is a type of second-order difference equation with parameterized coefficients. We then use this second-order difference equation to provide a bound. However, the core difficulty is that this difference equation is difficult to analyze when the coefficients are parameterized. Due to the space limit, we believe the convergence analysis of more advanced local solvers, such as LocalCH and LocalSOR with optimal parameters, can be treated as independent work.

---

**Q3. The effectiveness of local bounds compared with existing local bounds. (Reviewer kfsj)**

**A:** Our bound is effective. By a refined analysis, the actual inequality is $ \overline{ \operatorname {vol} } (\mathcal{S} _T ) / \overline \gamma _T < 1 / \epsilon$  as long as $T \geq 1$. The inequality used in Line 664 $(\leq)$ is actually $(<)$, i.e., $\sum _{i=1}^{ |\mathcal S_t|} r _{u_i} / ||\boldsymbol{r}^{(0)} ||_1 < \sum _{i=1}^{ |\mathcal S_t|} r _{u_i} / ||\boldsymbol{r}^{(t)} ||_1$ as long as $t \geq 1$. This strict inequality is achievable as long as the algorithm runs at least for two iterations. The reason is that the monotonicity property of $|| \boldsymbol r^{(t)} ||_1$ is actually $|| \boldsymbol r^{(0)} ||_1 < || \boldsymbol r^{(1)} ||_1 <\cdots < ||  \boldsymbol r^{(t)} ||_1 < \cdots$ where there are at least some magnitudes that are moved from $\boldsymbol r^{(t)}$ to $\boldsymbol x^{(t)}$ per-iteration. Therefore, in the worst case, the improvement of our bound is significant if the factor $1/\epsilon$ is the main concern. We conduct a simple experiment to compare these two quantities. **Please check our attached PDF file to see the experimental explanations.**

---

**Q4. Comments on the performance of local solvers over different graph structures. (Reviewer JA1M, X7w6)**

**A:** Our local algorithms do not depend on graph types. A fundamental assumption of our paper is that the computed diffusion vectors are highly localizable, measured by the participation ratio. As demonstrated in Figure 1, almost all diffusion vectors have low participation ratios collected from 18 real-world graphs. It is worth studying whether high-participation ratio diffusion vectors exist in real-world graphs. Interestingly, when the participation ratio is high, local solvers still offer some level of speedup compared to global ones. To verify this, we conducted a simple experiment where the graphs are grid graphs, and the diffusion vectors have high participation ratios. The results indicate that local solvers are more efficient than global ones, even when the participation ratios are high. **Please check our attached PDF file to see the detailed experimental results.**

To summarize,  for the first time, we propose a novel framework for designing efficient local solvers that could be applied to many graph diffusion equations (GDEs). These GDEs solvers can be applied to local clustering, training graph neural networks improving the efficiency of these algorithms.

---
**References**

- [1] Fountoulakis, K., Roosta-Khorasani, F., Shun, J., Cheng, X., & Mahoney, M. W. (2019). Variational perspective on local graph clustering. Mathematical Programming, 174, 553-573.

- [2] Bojchevski, Aleksandar, Johannes Gasteiger, Bryan Perozzi, Amol Kapoor, Martin Blais, Benedek Rózemberczki, Michal Lukasik, and Stephan Günnemann. "Scaling graph neural networks with approximate PageRank." In Proceedings of the 26th ACM SIGKDD International Conference on Knowledge Discovery & Data Mining, pp. 2464-2473. 2020.

---

### Author Response · Authors · 2024-08-14
**We thank all reviewers for their time and efforting, we will keep improving our submission.**

We sincerely thank all reviewers for their time and effort in the discussion and for helping us; we will keep improving our submission.

---

### Decision · Program_Chairs · 2024-09-25

**Decision:**

Accept (poster)

**Comment:**

This paper introduces a set of efficient methods for approximating graph diffusion vectors. Their approach achieves significant acceleration when implemented on GPUs. Empirical results demonstrate that these new methods can achieve speedups of up to one hundred times compared to their global counterparts.

Although the score is borderline, I find this contribution meaningful since it can potentially have an impact in speeding up computation which appears in the industry sector.